# An Asymptotically Optimal Algorithm for the Convex Hull Membership Problem

**Gang Qiao**                                                          *qiaogang@umich.edu*
*Department of Statistics*
*University of Michigan, Ann Arbor*

**Ambuj Tewari**                                                       *tewaria@umich.edu*
*Department of Statistics*
*University of Michigan, Ann Arbor*

**Reviewed on OpenReview:** *https://openreview.net/forum?id=r8eAwBMtlN*

## Abstract

We study the convex hull membership (CHM) problem in the pure exploration setting where one aims to efficiently and accurately determine if a given point lies in the convex hull of means of a finite set of distributions. We give a complete characterization of the sample complexity of the CHM problem in the one-dimensional case. We present the first asymptotically optimal algorithm called Thompson-CHM, whose modular design consists of a stopping rule and a sampling rule. In addition, we extend the algorithm to settings that generalize several important problems in the multi-armed bandit literature. Furthermore, we discuss the extension of Thompson-CHM to higher dimensions. Finally, we provide numerical experiments to demonstrate the empirical behavior of the algorithm matches our theoretical results for realistic time horizons.

## 1 Introduction

The multi-armed bandit (MAB) problem is a fundamental problem in sequential decision making where an agent is required to make a series of decisions to pull an arm of a $K$ slot machine in order to maximize the total reward. Each of the arms is associated with a fixed but unknown probability distribution (Auer et al., 2002; Lai et al., 1985). An enormous literature has accumulated over the past decades on the MAB problem, such as clinical trials and drug testing (Bastani & Bayati, 2020; Durand et al., 2018), recommendation system and online advertising (Bouneffouf et al., 2012; 2014; Nguyen, 2021; Tang et al., 2013; Zhou et al., 2017), information retrieval (Bounefouf et al., 2013; Losada et al., 2017), and finance (Huo & Fu, 2017; Misra et al., 2019; Mueller et al., 2019; Shen et al., 2015). The MAB problem was first studied theoretically in the seminal work (Robbins, 1952) and followed by a vast line of work in two canonical settings: regret minimization (Agrawal & Goyal, 2013; Auer, 2002; Auer et al., 2002; Chapelle & Li, 2011; Chu et al., 2011; Dudik et al., 2011; Langford & Zhang, 2007; Li et al., 2010; Srinivas et al., 2009; Valko et al., 2013) and pure exploration (Chen et al., 2017; Garivier & Kaufmann, 2016; Locatelli et al., 2016; Russo, 2016).

In this paper, we study the convex hull membership (CHM) problem in a pure exploration setting: testing whether a fixed target point lies in the convex hull of the unknown mean vectors of $K$ distributions as efficiently and accurately as possible. Following the multi-armed sampling model, where each distribution corresponds to an arm, and pulling an arm reveals an independent sample from the distribution. Unlike classical reward-maximizing bandit setting, our goal is to resolve a geometric decision problem under limited stochastic feedback. Pure exploration problems are usually studies in one of two settings: fixed-confidence of success or fixed-budget of samples. We work in the former. The usual non-stochastic version of the CHM problem is well studied in Filippozzi et al. (2023) and has attracted significant attention in different scientific areas and proven its crucial applications in image processing (Jayaram & Fleyeh, 2016; Yang &

Cohen, 1999), robot motion planning (Lengyel et al., 1990; Streinu, 2000) and pattern recognition (Katzin, 2018; Roy et al., 2008).

The stochastic CHM problem arises in important applications including fairness (Martinez et al., 2020) and multi-task learning (Lin et al., 2019), where we consider the instance to determine whether a given point $\boldsymbol{\theta}^* \in \Theta \subset \mathbb{R}^d$ lies on the Pareto frontier of a collection of $m$ objective functions $(F_1, \cdots, F_m)$. A point $\boldsymbol{\theta}^*$ is said to be Pareto optimal if no other $\boldsymbol{\theta} \in \Theta$ exists such that $F_i(\boldsymbol{\theta}) \leq F_i(\boldsymbol{\theta}^*)$ for all $i \in [m]$, with strict inequality in at least one coordinate. In is well known that, under differentiability, any Pareto optimal point satisfies a first-order condition: there exists a convex combination $(\lambda_1, \cdots, \lambda_m)$ with summation equal to 1 such that $\sum_{i=1}^m \lambda_i \nabla F_i(\boldsymbol{\theta}^*) = \mathbf{0}$, or equivalently, $\mathbf{0}_d \in \text{Conv}(\{\nabla F_i(\boldsymbol{\theta}^*)\}_{i=1}^m)$. In a multi-task learning setup, $\nabla F_i(\boldsymbol{\theta}) = \mathbb{E}_{P_i}[\nabla l_i(\boldsymbol{\theta})]$ where $P_i$'s and $l_i$'s are the underlying distributions and loss functions of the $i$-th task for $i \in [m]$. Since $P_i$'s are unknown, we utilize the empirical version of $\nabla F_i$ which follows distributions with different means and fits in our stochastic CHM setting. Nevertheless, there is almost no literature on the *stochastic* convex hull membership problem where we have to sample in order to estimate the positions of the means. Recently, Niss et al. (2022) provided the first theoretical bounds for the CHM problem. Unfortunately, their results have significant gaps between the upper and lower complexity bounds. To the best of our knowledge, this fundamental primitive of developing the complexity bounds and an (asymptotically) optimal algorithm for the CHM problem remains open in the literature before this work.

To tackle the aforementioned problem, we introduce Thompson-CHM, a Thompson-Sampling-based algorithm that has asymptotic sample complexity matching the information-theoretic lower bound proved in Garivier & Kaufmann (2016). The sample complexity lower bound is modeled as a function of the characteristic time (Garivier & Kaufmann, 2016), which can be captured by the value of a zero-sum pure exploration game between two players (Chernoff, 1959; Degenne et al., 2019). As discussed in Section 4, any successful pure exploration player needs to solve this pure exploration game, and therefore, the intuition behind the game is essential to our algorithm design. The design of the strategy to match the lower bound is based on the individual confidence interval for each of the $K$ arms so that any algorithm using this stopping time can ensure an output of a correct decision with high probability (at least $1 - \delta$) no matter what sampling rule the algorithm applies.

We remark that Kaufmann et al. (2018) first proposed an active sequential testing procedure to study the lowest mean of a finite set of distributions, and provide a conditional modification of the popular heuristic Thompson Sampling (named as *Murphy Sampling*) to tackle the limitations of the Lower Confidence Bound algorithm (LCB) and standard Thompson Sampling in different settings. However, *a major challenge in extending Thompson Sampling to our CHM problem is to study the extreme means (largest and lowest mean in one-dimensional setting) simultaneously*. To tackle this challenge, we borrow a two-arm sampling construction proposed in the Best-Arm Identification setting. Russo (2016) pointed out that Thompson Sampling can have a poor asymptotic performance and this defect can be improved by a top-two arm sampling modification to prevent the algorithm from sampling the arm of interest too frequently. This modification automatically controls the measurement effort of each arm and ensures that the long-term asymptotic behavior is closely linked to the optimal allocation of the algorithm. *To the best of our knowledge, conditional Thompson sampling and two-arm sampling have never been combined before.*

Our novel sampling rule is independent of the confidence parameter $\delta$ and ensures the sampled proportion of each arm asymptotically matches the estimated-best allocation design derived by the pure exploration game in the one-dimensional setting. Therefore, it automatically adapts exploration for both feasible and infeasible cases in the CHM problem. We provide a theoretical analysis of the asymptotic optimality and extend it to two more important settings: interval CHM problem (identifying if an interval $(\gamma^-, \gamma^+)$ intersects with the convex hull of the means of $K$ arms) and the $d$-dimensional CHM problem when $d \geq 2$. The first extension generalizes the CHM problem and reproduces the state-of-art results for several important MAB problems in the literature, including thresholding bandit (Locatelli et al., 2016) and sequential test for lowest mean (Kaufmann et al., 2018). Moreover, the stochastic CHM problem in $d$-dimensional setting has several important applications but its complete solution remains open.

To highlight and summarize our results, our contribution in this work is threefold:

- We prove the information-theoretical lower bound on the sample complexity of the one-dimensional convex hull membership (CHM) problem and reveal an oracle allocation of different arms for algorithm design.

- We introduce a novel Thompson-Sampling-based algorithm that automatically adapts the right exploration and oracle allocation for both feasible and infeasible cases and we rigorously prove the algorithm is asymptotically optimal and its complexity exactly matches the theoretical lower bound.

- Our final contribution is two important extensions of the Thompson-CHM algorithm. First, we extend the algorithm to the one-dimensional interval CHM problem by presenting the sample complexity bounds and the analogous asymptotically optimal algorithm, and discuss how this extension generalizes several fundamental BAI problems in the literature. We also investigate the potential extension to the $d$-dimensional CHM problem ($d \geq 2$) by showing the sample complexity bound which shares the same behavior as the one-dimensional case, and defer more details including the variant of the Thompson-CHM algorithm to the appendix.

## 2 Related Work

In this section, we briefly discuss some works and applications that motivate our work and are closely related to the convex hull membership problem in the literature.

**Thresholding Bandits:** One closely related previous work is a popular combinatorial pure exploration bandit problem known as the thresholding bandit problem where the learner's objective is to find the set of arms whose means are above a threshold. It was first introduced in Chen et al. (2014) and has been extensively studied in both fixed-confidence and fixed-budget settings (Chen et al., 2014; Garivier et al., 2017; Kano et al., 2019; Locatelli et al., 2016; Tao et al., 2019). Compared to the thresholding bandit problem, the convex hull membership problem only requires a boolean decision and needs the existence for both arms above and below the threshold to guarantee feasibility. A naive approach using the thresholding bandit problem to solve the CHM problem is to find the set of arms with means above and below the threshold by applying a thresholding bandit algorithm twice, and use the results to build a conclusion on if the threshold lies in the convex hull of the set of the arm means. Compared to the proposed Thompson-CHM algorithm, this two-step procedure is sub-optimal and expends unnecessary samples to determine the true sets of arms with means above and below the threshold.

**Fair Sampling and Minimax Pareto Fairness:** A recent series of works on fairness sampling and minimax Pareto fairness (Abernethy et al., 2020; Anahideh et al., 2022; Martinez et al., 2020; Nargesian et al., 2021) share similar frameworks with the fair data sampling procedure that is related to the CHM problem. As discussed in Niss et al. (2022), the main challenge of fair data sampling is to collect data of desired distribution requirements, therefore it reveals an appropriate representation of majority and minority groups in the data. In Anahideh et al. (2022), authors propose a fair active learning framework to balance the trade-off between model accuracy and fairness, in order to avoid discrimination in machine learning models. In Martinez et al. (2020), group fairness is formulated as a multi-objective optimization problem and proposes conditions for the classifier to be Pareto-efficient and achieve minimax risk, which is closely related to the stochastic CHM setting.

## 3 Problem Setup and Formulation

We define the problem of efficiently and accurately identifying if a given point lies in (or if a given interval/set intersects with, respectively) the convex hull of means of $K$ probability distributions $\nu_1, \cdots, \nu_K$ in dimension $d$ based on their stochastic sequential samples as the $d$-dimensional convex hull membership (*d-dim* CHM) problem. In this paper, we start with the one-dimensional setting where the probability distributions are in the canonical one-dimensional exponential family. In the canonical one-dimensional exponential family, the marginal distribution of a value $x$ given an unknown parameter $\theta \in \mathbb{R}$ takes the form

$$P(x|\theta) = h(x) \exp\{\eta(x)\theta - A(\theta)\}$$

where $h(x)$, $\eta(x)$ and $A(\theta)$ are known functions.

Throughout the paper, we denote by $\boldsymbol{\mu} = (\mu_1, \cdots, \mu_K)$ the vector of unknown *true* means of the distributions $\nu_1, \cdots, \nu_K$, and $\boldsymbol{\lambda}$ will be used as possible alternatives of the mean vector. The Kullback-Leibler divergence is a standard measure of how one probability distribution $P$ differs from another $Q$ with the form $\sum_x P(x) \ln(P(x)/Q(x))$. For the canonical one-dimensional exponential family, it induces a bijection between the natural parameter and the mean parameter, and we define the Kullback-Leibler divergence of two distributions with means $\mu_1$ and $\mu_2$ as a function $d : (\mu_1, \mu_2) \to \mathbb{R}^+$. Let $\text{Conv}(\boldsymbol{\mu}) = \text{Conv}(\mu_1, \cdots, \mu_K)$ denote the convex hull of the mean vector, which is the smallest convex set that contains all the means $\mu_1, \cdots, \mu_K$. At each time $t = 1, 2, \cdots$, a decision maker chooses one arm $A_t \in \{1, \cdots, K\}$ and independently draws a reward from distribution $X_{t, A_t} \sim \nu_{A_t}$. Let $\mathcal{F}_t$ denote the sigma algebra generated by $(A_1, X_{1, A_1}, \cdots, A_t, X_{t, A_t})$. We aim to design a sequential hypothesis testing procedure that consists of a $\mathcal{F}_{t-1}$-measurable *sampling policy* $\pi_t$, a *stopping rule* $\tau$ with respect to $\mathcal{F}_t$, and a $\mathcal{F}_\tau$-measurable *decision rule* $I_{\boldsymbol{\pi}}(\gamma) \in \{\text{feasible}, \text{infeasible}\}$.

We now state the formal definition of feasible and infeasible cases.

**Definition 3.1.** (feasibility and infeasibility) Given $\boldsymbol{\mu} = (\mu_1, \cdots, \mu_K)$, where $\mu_i \in \mathbb{R}^d$ for $i = 1, \cdots, K$. For any set $S$, the problem defined above is $S$-feasible if the set $S \cap \text{Conv}(\boldsymbol{\mu}) \neq \emptyset$, otherwise, the problem is $S$-infeasible. When the set only contains a single element $S = \{\gamma\}$, the problem is simply called $\gamma$-feasible and $\gamma$-infeasible, respectively.

In the one-dimensional case, given a threshold $\gamma \in \mathbb{R}$, our objective is to identify whether the unknown mean vector $\boldsymbol{\mu}$ is $\gamma$-feasible, which is equivalent to determining if $\gamma$ lies in the closed interval between the smallest mean reward $I_*(\boldsymbol{\mu}) = \text{argmin}_{1 \leq i \leq K} \mu_i$ and the largest mean reward $I^*(\boldsymbol{\mu}) = \text{argmax}_{1 \leq i \leq K} \mu_i$ based on the sequential observations while minimizing the expected stopping time $\tau$ and maximizing the probability to correctly identify the result. For simplicity, we assume the threshold $\gamma$ (and the extreme points of the set $S$, respectively) does not equal to $\mu_{I_*(\boldsymbol{\mu})}$ and $\mu_{I^*(\boldsymbol{\mu})}$. This aims to avoid infinite samples to distinguish the maximum and minimum means of the distributions that are too close to the threshold. This assumption can be easily relaxed by introducing a "precision" term $\varepsilon$ to identify an $\varepsilon$-optimal design instead (Locatelli et al., 2016; Russo, 2016). Additionally, we further assume the extreme points (or the vertices) of the convex hull $\text{Conv}(\boldsymbol{\mu})$ are unique.

In the literature, two distinct settings have been extensively studied. In the *fixed-confidence setting*, given a fixed confidence parameter $\delta \in (0, 1)$, the forecaster aims for a strategy that achieves the confidence $\delta$ about the quality of the decision rule while minimizing the sample needed, and in the *fixed-budget setting*, the number of exploration rounds is fixed, and the forecaster tries to maximize the probability of making the right decision. We will focus on the *fixed-confidence setting* in this paper and introduce the $\delta$-correct strategy.

**Definition 3.2.** ($\delta$-correctness) Let $\mathcal{D}$ be a set of distributions on $\mathbb{R}^d$. Given $\delta \in (0, 1)$, we call an identification strategy $\delta$-correct on the problem class $\boldsymbol{\nu} \in \mathcal{D}^K$ if with probability at least $1 - \delta$, the strategy returns the correct underlying case in a finite expected stopping time, i.e., $\mathbb{P}(\mathbb{E}[\tau] \leq \infty) = 1$, and when $\boldsymbol{\mu}$ is feasible, $\mathbb{P}(I_{\boldsymbol{\pi}}(\gamma) = \text{feasible}) \geq 1 - \delta$, otherwise $\mathbb{P}(I_{\boldsymbol{\pi}}(\gamma) = \text{infeasible}) \geq 1 - \delta$, here $I_{\boldsymbol{\pi}}(\gamma)$ is the decision rule of the strategy.

Before continuing, we pause to introduce some further notations here. We let $N_a(t) = \sum_{s=1}^t \mathbb{1}\{A_s = a\}$ be the number of selections of arm $a$ up to round $t$, and $S_a(t) = \sum_{s=1}^t X_s \mathbb{1}\{A_s = a\}$ be the sum of the gathered observations from that arm and $\hat{\mu}_a(t) = S_a(t)/N_a(t)$ be their empirical mean.

# 4 A General Lower Bound

In this section, we extend the general information-theoretical sample complexity lower bound proved in Garivier & Kaufmann (2016) to work for the one-dimensional convex hull membership problem.

We define $\text{Alt}(\boldsymbol{\mu})$ to be the set of bandit models where the identification result is different from that in $\boldsymbol{\mu}$, and $\Delta = \{\boldsymbol{w} = (w_1, \cdots, w_K) \in \mathbb{R}_+^K | w_1 + \cdots + w_K = 1\}$ is a probability simplex of dimension $K$. The following bound was proved by Garivier & Kaufmann (2016) that $\mathbb{E}_{\boldsymbol{\mu}}[\tau] \geq T^*(\boldsymbol{\mu})\text{kl}(\delta, 1 - \delta)$, where

$\mathrm{kl}(x, y) = x \ln(\frac{x}{y}) + (1 - x) \ln(\frac{1-x}{1-y})$ denotes the Kullback-Leibler divergence in the binary reward case, and

$$T^*(\boldsymbol{\mu})^{-1} = \sup_{\boldsymbol{w} \in \Delta} \inf_{\boldsymbol{\lambda} \in \mathrm{Alt}(\boldsymbol{\mu})} \sum_a w_a d(\mu_a, \lambda_a).$$

Note that $\mathrm{kl}(\delta, 1 - \delta) \sim \ln(1/\delta)$ as $\delta \to 0$, the lower bound above directly implies $\liminf_{\delta \to 0} \frac{\mathbb{E}_{\boldsymbol{\mu}}[\tau]}{\ln(1/\delta)} \geq T^*(\boldsymbol{\mu})$. This max-min problem was first discussed in the seminal work by Chernoff (1959), and the value of $T^*(\boldsymbol{\mu})^{-1}$ can be viewed as the value of a zero-sum simultaneous-move pure exploration game between two players. The player SUP aims to choose an optimal proportion of allocations $\boldsymbol{w}$ as a mixed strategy, and the adversary player INF tries to choose the worst-case alternative arm means that is hard to distinguish from the underlying truth to mislead SUP to an incorrect answer.

This general information-theoretic bound was established to analyze the sample complexity of the Best-Arm Identification problem (Garivier & Kaufmann, 2016), and was studied in different settings (Degenne et al., 2019; 2020) along with its popular variant that tackles pure exploration bandit problems with multiple correct answers (Degenne & Koolen, 2019). To match this general lower bound, the sampling proportion $\mathbb{E}[N_{\tau_\delta}] / \mathbb{E}_{\boldsymbol{\mu}}[\tau_\delta]$ must converge to the minimizer $\boldsymbol{w}^* \in \Delta$ of the pure exploration game as $\delta \to 0$. This intuition inspires works on different sampling rules and their corresponding threshold functions $\beta(t, \delta)$ to ensure correct recommendation with high probability (at least $1 - \delta$) (Degenne et al., 2019; Kaufmann & Koolen, 2021), and novel sampling rules to match the lower bound $\boldsymbol{N}(t)/t \to \boldsymbol{w}^*$, where $\boldsymbol{N}(t)$ is the vector of selection counts (Kaufmann et al., 2018). With these considerations in mind, we can establish the sample complexity bound and the asymptotically optimal algorithm for the CHM problem. Specifically, following Degenne et al. (2020), we say that a $\delta$-correct algorithm is *asymptotically optimal* if for all $\boldsymbol{\mu}$, $\limsup_{\delta \to 0} \frac{\mathbb{E}_{\boldsymbol{\mu}}[\tau]}{\ln(1/\delta)} \leq T^*(\boldsymbol{\mu})$.

Without loss of generality, in the one-dimensional CHM problem, we assume that $\mu_1 < \mu_2 \leq \mu_3 \leq \cdots \leq \mu_{K-1} < \mu_K$. The strict inequalities come from the aforementioned assumption of unique extreme points of $\mathrm{Conv}\{\boldsymbol{\mu}\}$. We have the following lower bound of any $\delta$-correct algorithm. The proof is provided in the appendix.

**Theorem 1.** *Given a threshold $\gamma \in \mathbb{R}$, the expected sample complexity $\mathbb{E}_{\boldsymbol{\mu}}[\tau]$ of any $\delta$-correct 1-dimensional CHM strategy satisfies $\liminf_{\delta \to 0} \frac{\mathbb{E}_{\boldsymbol{\mu}}[\tau]}{\ln(1/\delta)} \geq T^*(\boldsymbol{\mu})$, where*

$$T^*(\boldsymbol{\mu}) = \begin{cases} \frac{1}{d(\mu_1, \gamma)} + \frac{1}{d(\mu_K, \gamma)} & \gamma \in Conv\{\boldsymbol{\mu}\} \\ \sum_{1 \leq i \leq K} \frac{1}{d(\mu_i, \gamma)} & \gamma \notin Conv\{\boldsymbol{\mu}\} \end{cases},$$

*and*

$$w_a^*(\boldsymbol{\mu}) = \begin{cases} \frac{\frac{1}{d(\mu_a, \gamma)}}{\sum_{i \in \{1, K\}} \frac{1}{d(\mu_i, \gamma)}} \mathbb{1}_{\{a \in \{1, K\}\}} & \gamma \in Conv\{\boldsymbol{\mu}\} \\ \frac{\frac{1}{d(\mu_a, \gamma)}}{\sum_{1 \leq i \leq K} \frac{1}{d(\mu_i, \gamma)}} & \gamma \notin Conv\{\boldsymbol{\mu}\} \end{cases}.$$

Surprisingly, the characteristic time and oracle weights that match the general information-theoretic sample complexity show completely different behaviors in feasible and infeasible cases. In the feasible case where $\tau$ lies in the convex hull $\mathrm{Conv}(\boldsymbol{\mu})$, the algorithm should only sample the arms with minimum and maximum means, while in the infeasible case, the strategy should sample every single arm with specific fraction inversely proportional to the Kullback-Leibler divergence between its mean and the threshold $\gamma$. We remark that the previous work on sequentially testing and learning the lowest mean (Kaufmann et al., 2018) demonstrates a similar phenomenon. In essence, this commonality arises from the fact that the one-dimensional CHM problem generalizes the problem of learning the smallest mean (see section 6.1 for details).

## 5 Algorithm

In this section, we introduce an asymptotically optimal Thompson-Sampling-based algorithm for the one-dimensional CHM problem for a given threshold $\gamma \in \mathbb{R}$.

## 5.1 Stopping rule

From a learning point of view, the question of stopping at time $t$ is essentially a classical statistical problem: does the past collected information allow us to assess that the threshold $\gamma$ lies in or outside the convex hull set $\mathrm{Conv}(\boldsymbol{\mu})$ with risk at most $\delta$? Inspired by Kaufmann et al. (2018), a natural design of the stopping rule is to compare separately each arm to the threshold $\gamma$ and stop when either one arm lies significantly below $\gamma$ and one arm lies significantly above $\gamma$, or all arms lie significantly below $\gamma$, or all arms lie significantly above $\gamma$.

We denote $d^+(u,v) = d(u,v)\mathbb{1}\{u \le v\}$ and $d^-(u,v) = d(u,v)\mathbb{1}\{u \ge v\}$. We define the first stopping time $\tau_1$ when all arms lie significantly above $\gamma$:

$$\tau_1 = \inf\{t \in \mathbb{N}^+ | \forall a, N_a(t)d^-(\hat{\mu}_a(t), \gamma) \ge \mathrm{Thresh}(\delta, N_a(t))\}.$$

Similarly, we define the second stopping time $\tau_2$ when all arms lie significantly below $\gamma$:

$$\tau_2 = \inf\{t \in \mathbb{N}^+ | \forall a, N_a(t)d^+(\hat{\mu}_a(t), \gamma) \ge \mathrm{Thresh}(\delta, N_a(t))\}.$$

The third stopping time is when one arm is significantly below $\gamma$ and another arm lies significantly above $\gamma$:

$$\tau_3 = \inf\{t \in \mathbb{N}^+ | \exists a_1, a_2, \text{ such that } N_{a_1}(t)d^+(\hat{\mu}_{a_1}(t), \gamma) \ge \mathrm{Thresh}(\delta, N_{a_1}(t))$$
$$\text{and } N_{a_2}(t)d^-(\hat{\mu}_{a_2}(t), \gamma) \ge \mathrm{Thresh}(\delta, N_{a_2}(t))\}.$$

Here $\mathrm{Thresh}(\delta, N_a(t))$ is a threshold function to be specified later. Our algorithm stops if any of the three cases happen, i.e., it stops at $\tau = \min\{\tau_1, \tau_2, \tau_3\}$ and returns feasibility or infeasibility based on the case detected. The stopping rule and decision rule ensures that, when the threshold function $\mathrm{Thresh}(\delta, N_a(t))$ is carefully designed and the sampling rule guarantees the sampling allocation proportion converges to the solution $\boldsymbol{w}$ of the max-min problem, the algorithm Thompson-CHM is $\delta$-correct.

**Lemma 5.1.** *Let $\tau_\delta$ be a stopping rule satisfying $\tau_\delta \le \tau$. $\tau$ is a stopping rule whose threshold function $\mathrm{Thresh}(\delta, r)$ is non-decreasing in $r$ and satisfies the following: $\forall r \ge r_0$, $\mathrm{Thresh}(\delta, r) \le \ln(r/\delta) + o(\ln(1/\delta))$, then for any $\boldsymbol{\mu}$ and an anytime sampling strategy such that $\frac{N_t}{t} \to w^*(\boldsymbol{\mu})$, we have $\limsup_{\delta \to 0} \frac{\tau_\delta}{\ln(1/\delta)} \le T^*(\mu)$ almost surely.*

## 5.2 Sampling rule

Our contribution is a sampling rule that extends and generalizes a variant of Thompson sampling (called *Murphy Sampling*) introduced in Kaufmann et al. (2018) to the one-dimensional CHM problem that ensures the algorithm allocates the optimal proportion to each arm asymptotically, therefore guarantees the asymptotical optimality by Lemma 5.1. The sampling rule can automatically adapt the asymptotic optimality for both feasible cases and infeasible cases.

We denote by $\Pi_t = \mathbb{P}(\cdot|\mathcal{F}_t)$ the posterior distribution of the mean parameters after $t$ rounds. Inspired by Kaufmann et al. (2018) that introduces *Murphy Sampling* after Murphy's Law, as it performs some conditioning to the "worst event" to learn the smallest mean, we introduce Thompson-CHM (Algorithm 1) to tackle the one-dimensional CHM problem.

Note that the top-two Thompson Sampling conditions the standard Thompson Sampling on the event $\mathrm{argmax}\,\boldsymbol{\mu} \ne \mathrm{argmax}\,\boldsymbol{\theta}_t$ with pre-specified probability $\beta$ (Russo, 2016), and the Murphy Sampling conditions on $\min(\boldsymbol{\mu})$ below the threshold (Kaufmann et al., 2018). In contrast, Thompson-CHM conditions on the "feasibility" of the underlying mean vector $\boldsymbol{\mu}$ and in each round $t$, the algorithm proceeds to pull an arm in the sample $\boldsymbol{\theta}_t = (\theta_{t,1}, \cdots, \theta_{t,K})$ with largest or smallest mean based on the previous information $\mathcal{F}_{t-1}$. The posterior is computed explicitly using Bayes' rule with tractable priors (e.g. Beta or uniform). To implement the conditioning, we adopt reject sampling: we repeatedly sample from the unconditioned posterior until a sample satisfying feasibility is obtained. The next theorem guarantees that following this sampling procedure, the algorithm Thompson-CHM can ensure the sampling proportion of each arm converges to the optimal allocation $\boldsymbol{w}^*$ asymptotically, regardless of the position of $\gamma$ with respect to the convex hull $\mathrm{Conv}\{\boldsymbol{\mu}\}$. Therefore, we can conclude that the algorithm Thompson-CHM is asymptotically optimal in sample complexity.

---

**Algorithm 1** Thompson-CHM

---

**Input:** stopping rule $\tau$ with threshold function $\text{Thresh}(\delta, t)$, risk $\delta$, threshold $\gamma$, Bernoulli distribution parameter $\beta_t$.
**Output:** decision rule $I_\pi(\boldsymbol{\mu}) \in \{\text{feasible}, \text{infeasible}\}$
**for** $t = 1, \cdots$ **do**
  **if** stopping rule $\tau$ holds **then**
    **if** $\tau = \tau_3$ **then**
      **return** $I_\pi(\boldsymbol{\mu}) = \{\text{feasible}\}$
    **else**
      **return** $I_\pi(\boldsymbol{\mu}) = \{\text{infeasible}\}$
    **end if**
  **end if**
  Sample $\boldsymbol{\theta}_t = (\theta_{t,1}, \cdots, \theta_{t,K}) \sim \Pi_{t-1}(\cdot | \boldsymbol{\mu} \text{ feasible})$.
  Sample $B \sim \text{Bernoulli}(\beta_t)$
  **if** $B = 1$ **then**
    Play arm $A_t = \text{argmin}(\boldsymbol{\theta}_t)$
  **else**
    Play arm $A_t = \text{argmax}(\boldsymbol{\theta}_t)$
  **end if**
**end for**

---

**Theorem 2.** *If $\beta_t = \frac{d(\min \boldsymbol{\theta}_t, \gamma)^{-1}}{d(\min \boldsymbol{\theta}_t, \gamma)^{-1} + d(\max \boldsymbol{\theta}_t, \gamma)^{-1}}$, then the algorithm Thompson-CHM ensures that $\frac{N_t}{t} \to w^*(\boldsymbol{\mu})$ almost surely for any $\boldsymbol{\mu}$, and is $\delta$-correct for the CHM problem.*

We let $\psi_a(t)$ be the posterior probability of sampling arm $a$ at time $t$, i.e. $\psi_a(t) = \mathbb{P}(A_t = a | \mathcal{F}_{t-1})$, and define $\Psi_a(t)$ and $\bar{\psi}_a(t)$ as the summation and mean of $\psi_a(t)$ over time $t$.

For the feasible case, the first step of the sampling rule performs the same as Thompson Sampling, and the probability of drawing the first arm at time $t$ can be written as a weighted sum (with weights $\beta_t$ and $1 - \beta_t$) of the posterior probabilities that the first sample in $\boldsymbol{\theta}_t$ is the maximum and minimum. The asymptotic convergence of sample proportions $N_1(t)/t \to w_1^*(\boldsymbol{\mu})$ can be derived by the combination of facts that the former probability converges to 1 and $\beta_t$ converges to $w_1^*(\boldsymbol{\mu})$. The proof of $N_K(t)/t \to w_K^*(\boldsymbol{\mu})$ is symmetric.

For the infeasible case when the lowest mean is larger than threshold $\gamma$ or the largest mean is smaller than $\gamma$, the core idea of the proof is based on the following proposition, and the complete proofs of Theorem 1, Lemma 5.1, and Theorem 2 are deferred to the appendix.

**Proposition 5.2.** *(Simplified version of Lemma 12 of Russo (2016)) Consider any sampling rule, if for any arm $a \in [K]$ and all $c > 0$, $\sum_t \psi_a(t) \mathbb{1}\{\bar{\psi}_a(t) \geq w_a^* + c\} < \infty$, then $\bar{\psi}(t) \to \boldsymbol{w}^*$.*

The above result gives a sufficient condition in which $\bar{\psi}(t)$ converges to the optimal allocation $\boldsymbol{w}^*$, and implies that for any arm $a$ that meets $\bar{\psi}_a(t) \geq w_a^* + c$, the arm has been over-allocated compared to the optimal proportion $w_a^*$. Hence the total measurements the arm gets must be bounded in order to reduce towards $w_a^*$ for optimality. The rest of the proof is to establish the condition holds for Thompson-CHM algorithm. We develop the conclusion by showing that, if arm $a$ has been over-allocated compared to $w_a^*$, then $\Pi_t(\theta_{t,a} < \gamma < \theta_{t,b})$ is exponentially small compared to $\max_{a,b} \Pi_t(\theta_{t,a} < \gamma < \theta_{t,b})$. Based on the known result, for any open set $\tilde{\Theta} \subset \Theta$, the posterior concentrates at rate $\Pi_t(\tilde{\Theta}) \doteq \exp\left(-t \min_{\boldsymbol{\lambda} \in \tilde{\Theta}} \sum_a \bar{\psi}_a(t) d(\mu_a, \lambda_a)\right)$, where $x_t \doteq y_t$ means $\frac{1}{t} \ln \frac{x_t}{y_t} \to 0$. Combined with the properties of $T^*(\boldsymbol{\mu})$ in the pure exploration game and the concentration rate of the posterior, we can show that there exists $\delta' > 0$ such that,

$$\psi_a(t) \sim \frac{\Pi_t(\theta_{t,a} < \gamma < \theta_{t,b})}{\max_{a,b} \Pi_t(\theta_{t,a} < \gamma < \theta_{t,b})} \leq \exp(-t(\delta' + \varepsilon_t)),$$

where $\varepsilon_t$ is a sequence converging to 0. This implies for any arm $a$ such that $\bar{\psi}_a(t) \geq w_a^* + c$, $\psi_a(t)$ has an exponential decay rate, and Proposition 5.2 immediately yields $\bar{\psi}(t) \to \boldsymbol{w}^*$.

It is worth mentioning that one can tackle the one-dimensional CHM problem by first checking if $\gamma$ is smaller than the minimum mean and then checking if $\gamma$ is larger than the maximum mean. Using the results in Kaufmann et al. (2018), this strategy's sample complexity is at most two times the sample complexity stated in Theorem 1. However, this procedure has obvious drawbacks compared to our solution. First, this procedure does not generalize to higher dimensions since minimum and maximum means have no analogs in higher dimensions. Moreover, even in the one-dimensional case, this procedure incurs sub-optimality in its sample complexity in the infeasible case. By sequentially checking the one-sided setting twice, the arm that is farthest away from $\gamma$ will be sampled more than the optimal $\boldsymbol{w}^*(\boldsymbol{\mu})$ (and all other arms will be sampled less than $\boldsymbol{w}^*(\boldsymbol{\mu})$, respectively), especially when the arms are not spread out significantly. This demonstrates the sub-optimality of this easy solution as our main results indicate an algorithm matching the theoretical lower bound should follow the optimal allocation $\boldsymbol{w}^*(\boldsymbol{\mu})$. More details are discussed in the appendix.

## 6 Extensions of the Thompson-CHM Algorithm

### 6.1 Interval CHM Problem

In this section, we show that our results in Section 4 and Section 5 are fully generalizable to the interval feasibility setting, where our goal is to determine if the open set $(\gamma^-, \gamma^+)$ intersects with the convex hull set of $\boldsymbol{\mu}$. Here, we allow $\gamma^-$ to be $-\infty$ and $\gamma^+$ to be $+\infty$ for better generalization results.

#### 6.1.1 Asymptotic Optimality and the Algorithm

We build the first result on the general sample complexity lower bound.

**Theorem 3.** *Given thresholds* $-\infty \leq \gamma^- \leq \gamma^+ \leq +\infty$, *let* $(\gamma^*, \mu^*) = \operatorname{argmin}_{\gamma \in \{\gamma^-, \gamma^+\}, \mu \in \boldsymbol{\mu}} |\gamma - \mu|$. *The expected sample complexity* $\mathbb{E}_{\boldsymbol{\mu}}[\tau]$ *of any* $\delta$-*correct 1-dimensional CHM strategy satisfies* $\liminf_{\delta \to 0} \frac{\mathbb{E}_{\boldsymbol{\mu}}[\tau]}{\ln(1/\delta)} \geq T^*(\boldsymbol{\mu})$, *where*

$$T^*(\boldsymbol{\mu}) = \begin{cases} \frac{1}{d(\mu_1, \gamma^+)} + \frac{1}{d(\mu_K, \gamma^-)} & (\gamma^-, \gamma^+) \cap Conv\{\boldsymbol{\mu}\} \neq \emptyset \\ \sum_{1 \leq i \leq K} \frac{1}{d(\mu_i, \gamma^*)} & (\gamma^-, \gamma^+) \cap Conv\{\boldsymbol{\mu}\} = \emptyset \end{cases},$$

*and*

$$w_a^*(\boldsymbol{\mu}) = \begin{cases} \dfrac{\frac{1}{d(\mu_1, \gamma^+)} \mathbb{1}_{\{a=1\}} + \frac{1}{d(\mu_K, \gamma^-)} \mathbb{1}_{\{a=K\}}}{\frac{1}{d(\mu_1, \gamma^+)} + \frac{1}{d(\mu_K, \gamma^-)}} & (\gamma^-, \gamma^+) \cap Conv\{\boldsymbol{\mu}\} \neq \emptyset \\[4ex] \dfrac{\frac{1}{d(\mu_a, \gamma^*)}}{\sum_{1 \leq i \leq K} \frac{1}{d(\mu_i, \gamma^*)}} & (\gamma^-, \gamma^+) \cap Conv\{\boldsymbol{\mu}\} = \emptyset \end{cases}$$

The stopping rule is similar with minor adjustments. To be more specific, we again define the first stopping time $\tau_1$ when all arms lie significantly above $\gamma^+$:

$$\tau_1 = \inf\{t \in \mathbb{N}^+ | \forall a, N_a(t) d^-(\hat{\mu}_a(t), \gamma^+) \geq \text{Thresh}(\delta, N_a(t))\}.$$

Similarly, we define the second stopping time $\tau_2$ when all arms lie significantly below $\gamma^-$:

$$\tau_2 = \inf\{t \in \mathbb{N}^+ | \forall a, N_a(t) d^+(\hat{\mu}_a(t), \gamma^-) \geq \text{Thresh}(\delta, N_a(t))\}.$$

To identify the feasible case, the third stopping time is when one arm is significantly below $\gamma^+$ and another arm lies significantly above $\gamma^-$:

$$\tau_3 = \inf\{t \in \mathbb{N}^+ | \exists a_1, a_2, \text{ such that } N_{a_1}(t) d^+(\hat{\mu}_{a_1}(t), \gamma^+) \geq \text{Thresh}(\delta, N_{a_1}(t))$$
$$\text{and } N_{a_2}(t) d^-(\hat{\mu}_{a_2}(t), \gamma^-) \geq \text{Thresh}(\delta, N_{a_2}(t))\}.$$

Again, the algorithm stops if any of the three cases happens and $\tau = \min\{\tau_1, \tau_2, \tau_3\}$, and the following lemma ensures that the algorithm Thompson-CHM is $\delta$-correct in the interval feasibility framework.

**Lemma 6.1.** *Let $\tau_\delta$ be a stopping rule satisfying $\tau_\delta \leq \tau$. $\tau$ is a stopping rule whose threshold function Thresh$(\delta, r)$ is non-decreasing in $r$ and satisfies the following: $\forall r \geq r_0$, Thresh$(\delta, r) \leq \ln(r/\delta) + o(\ln(1/\delta))$, then for any $\boldsymbol{\mu}$ and an anytime sampling strategy such that $\frac{N_t}{t} \to w^*(\boldsymbol{\mu})$, we have $\limsup_{\delta \to 0} \frac{\tau_\delta}{\ln(1/\delta)} \leq T^*(\mu)$ almost surely.*

The sampling rule in the interval CHM problem remains the same and we have the next theorem.

**Theorem 4.** *If $\beta_t = \frac{d(\min \boldsymbol{\theta}_t, \gamma^+)^{-1}}{d(\min \boldsymbol{\theta}_t, \gamma^+)^{-1} + d(\max \boldsymbol{\theta}_t, \gamma^-)^{-1}}$, then the algorithm Thompson-CHM ensures that $\frac{N_t}{t} \to w^*(\boldsymbol{\mu})$ almost surely for any $\boldsymbol{\mu}$, and is $\delta$-correct for the CHM problem.*

### 6.1.2 Connections with Other State-of-art Results

We comment on some important connections of Section 6.1 with the previous state-of-art results in the MAB literature. Trivially, when $\gamma^- = \gamma^+$, we immediately derive the same results as the CHM problem, implying a direct generalization to the regular CHM problem. If we set $\gamma^- = -\infty$, the Bernoulli parameter $\beta_t$ becomes 0, and this reproduces the same Murphy Sampling results from the state-of-art sequential test paper for learning the minima mean (Kaufmann et al., 2018).

On the other hand, by setting $\gamma^+ = +\infty$, the interval CHM problem shares the same setup with the thresholding bandit problem with the threshold $\gamma^-$. In the infeasible case when $\gamma^-$ is larger than the largest mean in $\boldsymbol{\mu}$, testing if there exists an arm with a mean above the threshold is essentially equivalent to finding all arms with means above the threshold since to identify both questions, one needs to traverse all arms to conclude that the means of all arms are actually below the threshold, and our complexity bound exactly matches the state-of-art optimal bound of thresholding bandit (Locatelli et al., 2016). When $\mu$ is feasible, the CHM problem is strictly easier than the thresholding bandit, and our complexity is strictly smaller than the state-of-art result. Notably, the Thompson-CHM algorithm adapts both feasible and infeasible cases for the thresholding bandit problem without knowing any information on the threshold as a priori.

### 6.2 Convex hull membership problem in higher dimensions

We now investigate and discuss the extensions of the Thompson-CHM to $d$-dimensional setting where $d \geq 2$. Before proceeding, we define the vertices set (or extreme point set) Vert$(S)$ of a convex set $S$ to be the union of points that do not fall on any line segment connecting any two unique points in set $S$. The following theorem states that the lower bound for the CHM problem exhibits a shared behavior in all dimensions: *in the feasible case, the optimal strategy should only sample arms whose means are extreme points, and in the infeasible case, it should sample all arms.*

**Theorem 5.** *Let Vert$(Conv\{\boldsymbol{\mu}\}) = (\mu_{s_1}, \cdots, \mu_{s_m})$ be the vertices set of $Conv\{\boldsymbol{\mu}\}$. Given $\gamma \in \mathbb{R}^d$ where $2 \leq d < \infty$, the expected sample complexity $\mathbb{E}_{\boldsymbol{\mu}}[\tau]$ of any $\delta$-correct $d$-dimensional CHM strategy satisfies $\liminf_{\delta \to 0} \frac{\mathbb{E}_{\boldsymbol{\mu}}[\tau]}{\ln(1/\delta)} \geq T^*(\boldsymbol{\mu})$, where*

$$T^*(\boldsymbol{\mu}) = \begin{cases} f_0(\mu_{s_1}, \cdots, \mu_{s_m}, \gamma) & \gamma \in Conv\{\boldsymbol{\mu}\} \\ \sum_{1 \leq i \leq K} \frac{1}{d(\mu_i, \gamma)} & \gamma \notin Conv\{\boldsymbol{\mu}\} \end{cases},$$

*and*

$$w_a^*(\boldsymbol{\mu}) = \begin{cases} \sum_{i=1}^m f_i(\mu_{s_1}, \cdots, \mu_{s_m}, \gamma) \mathbb{1}_{\{a = s_i\}} & \gamma \in Conv\{\boldsymbol{\mu}\} \\ \frac{\frac{1}{d(\mu_a, \gamma)}}{\sum_{1 \leq i \leq K} \frac{1}{d(\mu_i, \gamma)}} & \gamma \notin Conv\{\boldsymbol{\mu}\} \end{cases}.$$

*Here $f_0, f_1, \cdots, f_m$ are non-negative real-value functions, and $\sum_{i=1}^m f_i(\mu_{s_1}, \cdots, \mu_{s_m}, \gamma) = 1$.*

Using Theorem 5, we can generalize the Thompson-CHM algorithm to higher dimensions by simply replacing the Bernoulli distribution with a categorical distribution with parameters $\beta_i = f_i(\mu_{s_1}, \cdots, \mu_{s_m}, \gamma)$ and assuming oracle access to the functions $f_i$ for $1 \leq i \leq k$. We discuss more details of the Thompson-CHM algorithm in $d$-dimensional setting $(d \geq 2)$ in the appendix.

# 7 Numerical results

The paper's main results are reflected in some numerical experiments in this section. We consider 7 Bernoulli bandits with means $\boldsymbol{\mu} = (0.1, 0.2, \cdots, 0.7)$ and $\delta = e^{-3}$, and we consider Beta(1,1) prior and different $\gamma$'s to compare the sample complexity and sample weights to the theoretical results in both feasible and infeasible cases. We use the threshold function developed in Kaufmann et al. (2018): $\mathrm{Thresh}(\delta, r) = \ln(1 + \ln(r)) + T(\ln(1/\delta))$ and $T : \mathbb{R}^+ \to \mathbb{R}^+$ is a function defined by $T(x) = 2h^{-1}\left(1 + \frac{h^{-1}(1+x) + \ln \zeta(2)}{2}\right)$, where $h(u) = u - \ln(u)$ for $u \geq 1$ and $\zeta(s) = \sum_{n=1}^{\infty} n^{-s}$. We also adopt the empirical implementation of $T(x)$ from Kaufmann et al. (2018) in our experiments. The property of the threshold function is verified in Kaufmann et al. (2018).

We first pick different values of $\gamma$ to compare the theoretical sample complexity and the sample complexity of Thompson-CHM in both feasible and infeasible cases. We choose $\gamma$ to be $(0.15, 0.25, 0.35, 0.45, 0.55, 0.65)$ for the feasible case and $(0.75, 0.8, 0.85, 0.9, 0.95)$ for the infeasible case. Figure 1 demonstrates the efficiency of the algorithm Thompson-CHM. In both feasible and infeasible cases, the sample complexity of Thompson-CHM matches the theoretical results proved in Theorem 1 well for realistic time horizons.

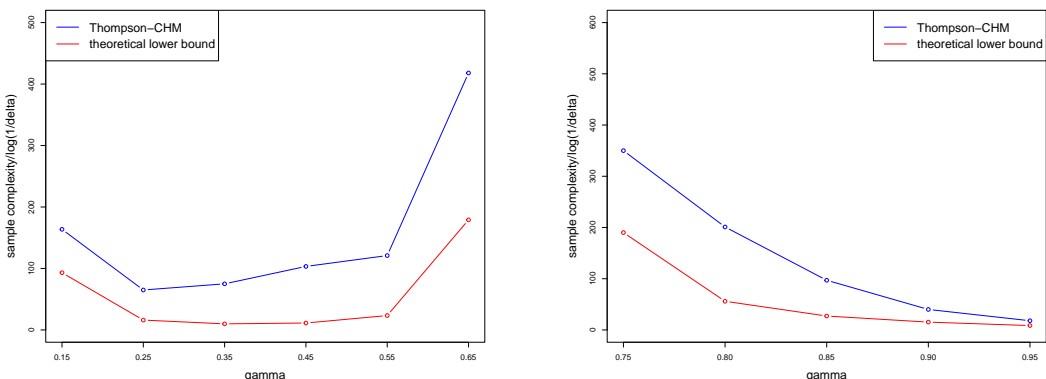

Figure 1: Sample complexity for different $\gamma$'s in feasible cases (left) and infeasible cases (right).

Figure 2 provides insights into the asymptotic convergence performance of the sampling proportions $N_a(\tau)/\tau$ in Thompson-CHM in both feasible and infeasible cases. In the feasible case when $\gamma = 0.25$, we note that Thompson-CHM spent the most fraction of time sampling the side arms (especially the minimum arm since $\gamma$ is much closer to the arm with minimum mean compared to the arm with maximum mean). In the infeasible case when $\gamma = 0.9$, we can observe that the sampling proportion of the algorithm almost matches the theoretical optimal $\boldsymbol{w}^*(\boldsymbol{\mu})$ in Theorem 1.

We further investigate how the sample complexity of Thompson-CHM scales with the confidence parameter $\delta$, as motivated by the logarithmic dependence predicted by our theoretical results. We choose $\gamma = 0.4$ for feasible case and $\gamma = 0.9$ for infeasible case, and a varying range of $-\log(\delta)$ to evenly range between 1 to 5. For each $\delta$, we run 100 independent trials and report the average sample complexity. As shown in Figure 3, the empirical sample complexity grows linearly with $\log(1/\delta)$, closely tracking the theoretical lower bound up to a constant factor. This supports the sharpness of our bound and confirms that the Thompson-CHM algorithm achieves asymptotic optimal sample complexity in terms of its $\delta$ dependence.

# 8 Discussion and conclusion

This work thoroughly investigates the convex hull membership (CHM) problem in the pure exploration setting. We propose a novel asymptotically optimal algorithm to tackle this problem, which we refer to as Thompson-CHM algorithm. The sampling rule combines the ideas of top-two Thompson sampling (Russo, 2016) and Murphy sampling (Kaufmann et al., 2018), and it can automatically guarantee the sampled

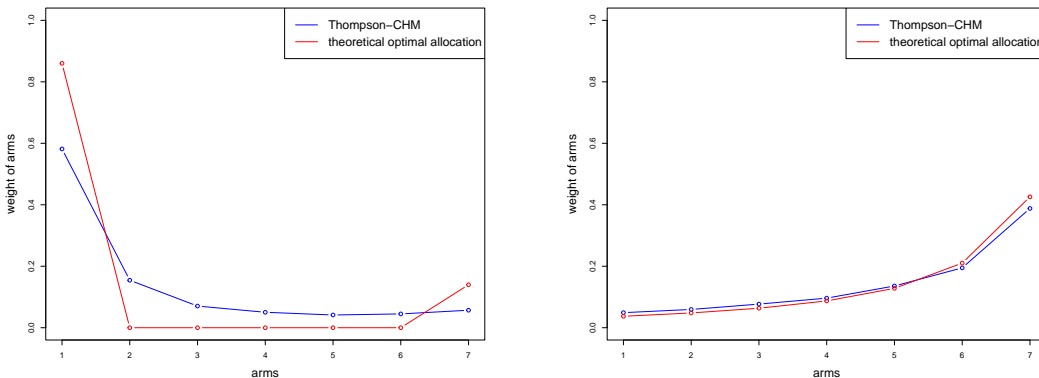

Figure 2: Empirical proportion of samples compared to optimal allocation $\boldsymbol{w}^*(\boldsymbol{\mu})$ in feasible cases (left) and infeasible cases (right) estimated using 100 repetitions.

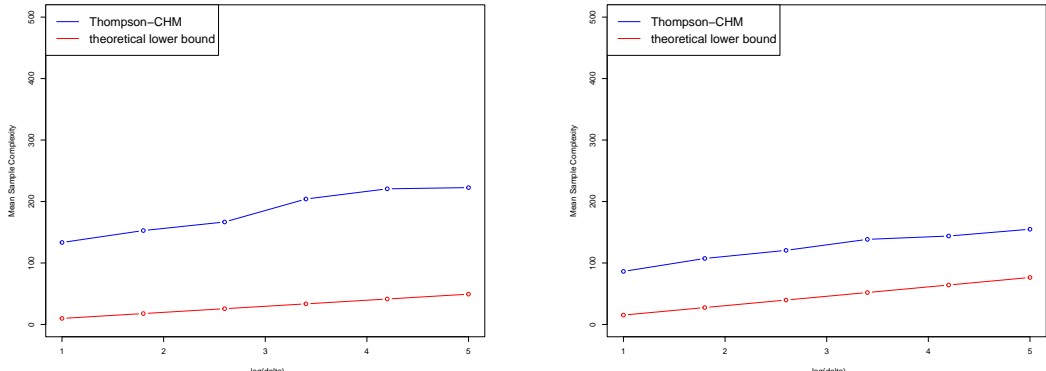

Figure 3: Sample complexity for different $\delta$'s in feasible cases (left) and infeasible cases (right).

proportion of each arm converges to the optimal allocation derived by the information-theoretical lower bound in the one-dimensional setting, regardless of relative position between the threshold and the arm mean set. Moreover, we extend our results to the interval CHM setting that generalizes some important MAB problems in the literature and investigate the extensions of the Thompson-CHM algorithm in higher dimensions.

Future work will attempt to derive a complete solution to $d$-dimensional CHM problems with broader settings when $d \geq 2$. We conjecture that the sample complexity bounds and the asymptotically optimal algorithm are identical to the one-dimensional case. The current theoretical results reveal challenges in the feasible $d$-dimensional setting due to the complex geometric structure in the "alternative" set. It would be interesting to fully understand the CHM problem in $d$-dimensional case when $d \geq 2$.

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

# A  Appendix and Proofs

## A.1  Proofs for one-dimensional $\gamma$-CHM problem

We provide proofs of Theorem 1, Lemma 5.1 and Theorem 2 in this Section. As discussed in Section 6.1.2, the interval CHM problem fully generalizes the regular CHM problem, and the analogous results in the interval CHM problem follow the exactly same proofs and can be derived by properly adjusting $\gamma$ to $\gamma^-$ and $\gamma^+$ in this section. Therefore, we only prove Theorem 1, Lemma 5.1 and Theorem 2 for the ease of extensions to the Gaussian bandit setting with unknown variances.

### A.1.1  Proof of Theorem 1

We recall $\mu_1 = \min\{\mu_1, \cdots, \mu_K\}$ and $\mu_K = \max\{\mu_1, \cdots, \mu_K\}$. In the one-dimensional case, the $\gamma$-CHM problem is to test if $\gamma \in \text{Conv}(\boldsymbol{\mu})$. For the feasible case,

$$\text{Alt}(\boldsymbol{\mu}) = \{\boldsymbol{\lambda}|I_\pi(\boldsymbol{\lambda}) = \text{infeasible}\} = \{\boldsymbol{\lambda}| \max_{1 \le i \le K} \lambda_i < \gamma \text{ or } \min_{1 \le i \le K} \lambda_i > \gamma\}.$$

Therefore,

$$
\begin{aligned}
T^*(\boldsymbol{\mu})^{-1} &= \max_{\boldsymbol{w} \in \Delta} \min_{\boldsymbol{\lambda} \in \mathrm{Alt}(\boldsymbol{\mu})} \sum_a w_a d(\mu_a, \lambda_a) \\
&= \max_{\boldsymbol{w} \in \Delta} \min \left( \sum_{a:\mu_a < \gamma} w_a d(\mu_a, \gamma), \sum_{a:\mu_a > \gamma} w_a d(\mu_a, \gamma) \right) \\
&= \max_{w_1 + w_K = 1} \min \left( w_1 d(\mu_1, \gamma), w_K d(\mu_K, \gamma) \right) \\
&= \frac{d(\mu_1, \gamma) d(\mu_K, \gamma)}{d(\mu_1, \gamma) + d(\mu_K, \gamma)} \\
&= \frac{1}{d(\mu_1, \gamma)^{-1} + d(\mu_K, \gamma)^{-1}} .
\end{aligned}
$$

From the derivation, we can see the optimization problem derives its optimal solution when the strategy only samples arms with maximum and minimum mean with proportions $w_1 = \frac{d(\mu_1, \gamma)^{-1}}{d(\mu_1,\gamma)^{-1} + d(\mu_K,\gamma)^{-1}}$ and $w_K = \frac{d(\mu_K, \gamma)^{-1}}{d(\mu_1,\gamma)^{-1} + d(\mu_K,\gamma)^{-1}}$. Now we consider the infeasible case. Without loss of generality, we assume $\mu_1 > \tau$ (the case when $\mu_K < \tau$ can be proved in the same way due to symmetry). In this case,

$$
\mathrm{Alt}(\boldsymbol{\mu}) = \{\boldsymbol{\lambda} | I_\pi(\boldsymbol{\lambda}) = \text{feasible}\} = \{\boldsymbol{\lambda} | \lambda_1 < \gamma < \lambda_K\}.
$$

and

$$
\begin{aligned}
T^*(\boldsymbol{\mu})^{-1} &= \max_{\boldsymbol{w} \in \Delta} \min_{\boldsymbol{\lambda} \in \mathrm{Alt}(\boldsymbol{\mu})} \sum_a w_a d(\mu_a, \lambda_a) \\
&= \max_{\boldsymbol{w} \in \Delta} \min_{1 \le a \le K} w_a d(\mu_a, \gamma) \\
&= \frac{1}{\sum_{1 \le a \le K} d(\mu_a, \gamma)^{-1}} .
\end{aligned}
$$

We can see from that in the infeasible case, the decision-maker should sample all arms, and for each arm $a \in \{1, \cdots, K\}$, it should be sampled with proportion $w_a = \frac{\frac{1}{d(\mu_a, \gamma)}}{\sum_{1 \le i \le K} \frac{1}{d(\mu_i, \gamma)}}$.

### A.1.2 Proof of Lemma 5.1

Now we proceed to prove Lemma 5.1. We will make use of the following proposition.

**Proposition A.1.** *(Lemma 22 of Kaufmann et al. (2016)) For every $\beta, \eta > 0$, if*

$$
x \ge \frac{1}{\beta} \ln \left( \frac{e \ln(1/\beta\eta)}{\beta\eta} \right),
$$

*then we have*

$$
\beta x \ge \ln \left( \frac{x}{\eta} \right) + o \left( \ln \left( \frac{1}{\eta} \right) \right).
$$

Now for the feasible case, on the event that

$$
\frac{N_1(t)}{t} \to w_1^*(\boldsymbol{\mu}), \quad \hat{\mu}_1(t) \to \mu_1,
$$

and

$$
\frac{N_K(t)}{t} \to w_K^*(\boldsymbol{\mu}), \quad \hat{\mu}_K(t) \to \mu_K,
$$

for any $\varepsilon > 0$, there exists $t_0 > 0$, such that for any $t > t_0$, we have

$$
N_1(t) d^+(\hat{\mu}_1(t), \gamma) \ge (1 - \varepsilon) t w_1^*(\boldsymbol{\mu}) d(\mu_1, \gamma),
$$

and

$$
N_K(t) d^-(\hat{\mu}_K(t), \gamma) \ge (1 - \varepsilon) t w_K^*(\boldsymbol{\mu}) d(\mu_K, \gamma).
$$

Following this,

$$\tau \leq \tau_3 \leq \inf\{t | N_1(t)d^+(\hat{\mu}_1(t),\gamma) \geq \text{Thresh}(\delta, N_1(t)) \text{ and } N_K(t)d^-(\hat{\mu}_K(t),\gamma) \geq \text{Thresh}(\delta, N_K(t))\}$$
$$\leq \inf\left\{t | N_K(t)d^+(\hat{\mu}_1(t),\gamma) \geq \text{Thresh}(\delta,t) \text{ and } N_K(t)d^+(\hat{\mu}_K(t),\gamma) \geq \text{Thresh}(\delta,t)\right\}$$
$$\leq \inf\left\{t | (1-\varepsilon)tw_1^*(\boldsymbol{\mu})d(\mu_1,\gamma) \geq \text{Thresh}(\delta,t) \text{ and } (1-\varepsilon)tw_K^*(\boldsymbol{\mu})d(\mu_K,\gamma) \geq \text{Thresh}(\delta,t)\right\}$$
$$\leq \inf\left\{t | (1-\varepsilon)t \min\left(w_1^*(\boldsymbol{\mu})d(\mu_1,\gamma), w_K^*(\boldsymbol{\mu})d(\mu_K,\gamma)\right) \geq \ln\left(\frac{t}{\delta}\right) + o\left(\ln\left(\frac{1}{\delta}\right)\right)\right\}$$
$$\leq \inf\left\{t | (1-\varepsilon)tT^*(\boldsymbol{\mu})^{-1} \geq \ln\left(\frac{t}{\delta}\right) + o\left(\ln\left(\frac{1}{\delta}\right)\right)\right\}.$$

Hence we have $\tau(1-\varepsilon)T^*(\boldsymbol{\mu})^{-1} \leq \ln\left(\frac{t}{\delta}\right) + o\left(\ln\left(\frac{1}{\delta}\right)\right)$. By setting $\beta = (1-\varepsilon)T^*(\boldsymbol{\mu})^{-1}$, $x = \tau$ and $\eta = \delta$, Proposition A.1 directly yields

$$\tau \leq \frac{1}{(1-\varepsilon)T^*(\boldsymbol{\mu})^{-1}} \ln\left(\frac{e\ln\left(\frac{1}{(1-\varepsilon)T^*(\boldsymbol{\mu})^{-1}\delta}\right)}{(1-\varepsilon)T^*(\boldsymbol{\mu})^{-1}\delta}\right)$$
$$\leq \frac{1}{(1-\varepsilon)T^*(\boldsymbol{\mu})^{-1}}\left(\ln\left(\frac{e}{(1-\varepsilon)T^*(\boldsymbol{\mu})^{-1}}\right) + \ln\left(\frac{1}{\delta}\right) + \ln\ln\left(\frac{1}{(1-\varepsilon)T^*(\boldsymbol{\mu})^{-1}\delta}\right)\right)$$
$$\leq \frac{1}{(1-\varepsilon)T^*(\boldsymbol{\mu})^{-1}} \ln\left(\frac{1}{\delta}\right) + o\left(\ln\left(\frac{1}{\delta}\right)\right)$$

Notice that $\varepsilon$ is arbitrary, we have

$$\limsup_{\delta\to 0} \frac{\tau}{\ln(1/\delta)} \leq T^*(\boldsymbol{\mu}).$$

For the infeasible case, WLOG we assume $\mu_1 > \gamma$ (proof of the symmetric case $\mu_K < \gamma$ is identical). On the event that for any arm $a \in \{1, \cdots, K\}$,

$$\frac{N_a(t)}{t} \to w_a^*(\boldsymbol{\mu}), \quad \hat{\mu}_a(t) \to \mu_a,$$

and for arbitrary $\varepsilon > 0$, similarly, there exists $t_0$, and for any $t > t_0$, we have the following inequality to hold:

$$N_a(t)d^-(\hat{\mu}_a(t),\gamma) \geq (1-\varepsilon)tw_a^*(\boldsymbol{\mu})d(\mu_a,\gamma).$$

Using a parallel statement of the feasible case,

$$\tau \leq \tau_3 \leq \inf\{t | \forall a, N_a(t)d(\hat{\mu}_a(t),\gamma) \geq \text{Thresh}(\delta, N_1(t))\}$$
$$\leq \inf\left\{t | \forall a, N_a(t)d^-(\hat{\mu}_a(t),\gamma) \geq \text{Thresh}(\delta,t)\right\}$$
$$\leq \inf\left\{t | \forall a, (1-\varepsilon)tw_a^*(\boldsymbol{\mu})d(\mu_a,\gamma) \geq \ln\left(\frac{t}{\delta}\right) + o\left(\ln\left(\frac{1}{\delta}\right)\right)\right\}$$
$$\leq \inf\left\{t | (1-\varepsilon)t \min_{1\leq a\leq K}\{w_a^*(\boldsymbol{\mu})d(\mu_a,\gamma)\} \geq \ln\left(\frac{t}{\delta}\right) + o\left(\ln\left(\frac{1}{\delta}\right)\right)\right\}$$
$$\leq \inf\left\{t | (1-\varepsilon)tT^*(\boldsymbol{\mu})^{-1} \geq \ln\left(\frac{t}{\delta}\right) + o\left(\ln\left(\frac{1}{\delta}\right)\right)\right\}.$$

Following the same statements and by applying Proposition A.1, we have

$$\tau \leq \frac{1}{(1-\varepsilon)T^*(\boldsymbol{\mu})^{-1}} \ln\left(\frac{1}{\delta}\right) + o\left(\ln\left(\frac{1}{\delta}\right)\right).$$

Thus, in the infeasible case,

$$\limsup_{\delta\to 0} \frac{\tau}{\ln(1/\delta)} \leq T^*(\boldsymbol{\mu})$$

also holds.

### A.1.3 Proof of Theorem 2

We again consider feasible and infeasible cases separately. We recall the following notations

$$\psi_a(t) = \mathbb{P}(A_t = a | \mathcal{F}_{t-1}), \quad \Psi_a(t) = \sum_{i=1}^{t} \psi_a(i), \quad \text{and} \quad \bar{\psi}_a(t) = \frac{1}{t} \Psi_a(t).$$

Our proof is based on a classic result (see Corollary 1 of Russo (2016)) that for any arm $a \in [K]$, if $\Psi_a(t) \to \infty$, then

$$\frac{w_a(t)}{\bar{\psi}_a(t)} = \frac{N_a(t)}{\Psi_a(t)} \to 1 \quad \text{a.s}$$

and the following result from Kaufmann et al. (2018).

**Proposition A.2.** *(Theorem 12 of Kaufmann et al. (2018)) Given a threshold $\gamma$, for any $\boldsymbol{\mu} = \{(\mu_1, \cdots, \mu_K) | \mu_1 \leq \cdots \leq \mu_K, \text{ and } \mu_1 < \gamma\}$. If we sequentially sample as follows: for any $t \in \mathbb{N}^+$, sample $\boldsymbol{\theta}_t \sim \Pi_{t-1}(\cdot | \min_{1 \leq i \leq K} \mu_i < \gamma)$, then play the arm $A_t$ with lowest mean in $\boldsymbol{\theta}_t$. Then the sampling procedure ensures that the sampling frequencies satisfy*

$$\frac{N_1(t)}{t} \to 1,$$

*and for any $2 \leq a \leq K$,*

$$\frac{N_a(t)}{t} \to 0$$

*almost surely.*

Back to our proof, now we consider the feasible case first. In this case, we have $\boldsymbol{\mu}$ with property $\mu_1 < \gamma < \mu_K$. For any $n \in \mathbb{N}^+$,

$$\psi_1(t) = \beta_t \Pi_t \left( \theta_{t,1} < \min_{j \neq 1} \theta_{t,j} | \boldsymbol{\mu} \text{ feasible} \right) + (1 - \beta_t) \Pi_t \left( \theta_{t,1} > \max_{j \neq 1} \theta_{t,j} | \boldsymbol{\mu} \text{ feasible} \right).$$

Notice that $\{\text{argmin}_a \theta_a = 1\}$ and $\{\mu_K > \gamma\}$ are independent events,

$$\Pi_t \left( \theta_{t,1} < \min_{j \neq 1} \theta_{t,j} | \boldsymbol{\mu} \text{ feasible} \right) \to 1 \quad a.s..$$

And $\Pi_t (\theta_{t,1} < \min_{j \neq 1} \theta_{t,j} | \boldsymbol{\mu} \text{ feasible}) + \Pi_t (\theta_{t,1} > \min_{j \neq 1} \theta_{t,j} | \boldsymbol{\mu} \text{ feasible}) \leq 1$ directly yields

$$\Pi_t \left( \theta_{t,1} > \max_{j \neq 1} \theta_{t,j} | \boldsymbol{\mu} \text{ feasible} \right) \to 0 \quad a.s..$$

Combine with the fact that $\beta_t \to \frac{d^{-1}(\mu_1, \gamma)}{d^{-1}(\mu_1, \gamma) + d^{-1}(\mu_K, \gamma)}$, we get $\frac{N_1(t)}{t} \to w_1^*(\boldsymbol{\mu})$. Similarly,

$$\psi_K(t) = \beta_t \Pi_t \left( \theta_{t,K} < \min_{j \neq 1} \theta_{t,j} | \boldsymbol{\mu} \text{ feasible} \right) + (1 - \beta_t) \Pi_t \left( \theta_{t,K} > \max_{j \neq 1} \theta_{t,j} | \boldsymbol{\mu} \text{ feasible} \right).$$

With the facts that $\Pi_t (\theta_{t,K} < \min_{j \neq 1} \theta_{t,j} | \boldsymbol{\mu} \text{ feasible}) \to 0$ and $\Pi_t (\theta_{t,K} > \max_{j \neq 1} \theta_{t,j} | \boldsymbol{\mu} \text{ feasible}) \to 1$ almost surely, this leads to $\frac{N_K(t)}{t} \to w_K^*(\boldsymbol{\mu})$. Notice that $w_1^*(\boldsymbol{\mu}) + w_K^*(\boldsymbol{\mu}) = 1$, we have shown that $\frac{\boldsymbol{N}(t)}{t} \to w^*(\boldsymbol{\mu})$ almost sure in the feasible case.

For the infeasible case, we use the following proposition.

**Proposition A.3.** *(Simplified version of Lemma of Russo (2016)) Consider any sampling rule, if for any arm $a \in [K]$ and all $c > 0$,*

$$\sum_t \psi_a(t) \mathbb{1}\{\bar{\psi}_a(t) \geq w_a^* + c\} < \infty,$$

*then $\bar{\psi}(t) \to \boldsymbol{w}^*$.*

By applying a similar proof strategy in Russo (2016) and Kaufmann et al. (2018), we aim to prove the precondition in Proposition 5.2. For any $a \in [K]$ and $c > 0$, consider any round $n$ where $\bar{\psi}_a(t) \geq w_a^* + c$, we have

$$\psi_a(t) = \beta_t \frac{\Pi_{t-1}(a = \operatorname{argmin}_i \theta_{t,i}, b = \operatorname{argmax}_i \theta_{t,i}, \min_i \theta_{t,i} < \gamma < \max_i \theta_{t,i})}{\Pi_{t-1}(\min_i \theta_{t,i} < \gamma < \max_i \theta_{t,i})}$$

$$+ (1 - \beta_t) \frac{\Pi_{t-1}(a = \operatorname{argmax}_i \theta_{t,i}, b = \operatorname{argmin}_i \theta_{t,i}, \min_i \theta_{t,i} < \gamma < \max_i \theta_{t,i})}{\Pi_{t-1}(\min_i \theta_{t,i} < \gamma < \max_i \theta_{t,i})}$$

$$\leq \beta_t \frac{\Pi_{t-1}(\theta_{t,a} < \gamma < \theta_{t,b})}{\max_{a,b} \Pi_{t-1}(\theta_{t,a} < \gamma < \theta_{t,b})} + (1 - \beta_t) \frac{\Pi_{t-1}(\theta_{t,b} < \gamma < \theta_{t,a})}{\max_{a,b} \Pi_{t-1}(\theta_{t,b} < \gamma < \theta_{t,a})}.$$

Following Russo (2016), recall we use $x_t \doteq y_t$ to denote that $t^{-1} \ln(x_t/y_t) \to 0$. Based on any known posterior concentration rate result (for example, Proposition 5 in Russo (2016)) that for any open set $\tilde{\Theta} \subset \Theta$, the posterior concentrates at the rate $\Pi_t(\tilde{\Theta}) \doteq \exp\left(-t \min_{\boldsymbol{\lambda} \in \tilde{\Theta}} \sum_a \bar{\psi}_a(t) d(\mu_a, \lambda_a)\right)$. Moreover, for any $a, b \in [K]$,

$$\Pi_t(\theta_{t,a} < \gamma < \theta_{t,b}) \doteq \exp\left(-t \min_{\boldsymbol{\theta}_t \text{ feasible}} \sum_a \bar{\psi}_a(t) d(\mu_a, \theta_{t,a})\right)$$

$$= \exp\left(-t \min\left(\sum_{a:\mu_a < \gamma} \bar{\psi}_a(t) d(\mu_a, \gamma), \sum_{a:\mu_a > \gamma} \bar{\psi}_a(t) d(\mu_a, \gamma)\right)\right).$$

This means, there is a sequence $\varepsilon_t \to 0$ such that for any $t$,

$$\Pi_t(\theta_a < \gamma < \theta_b) \in \exp\left(-t \left(\min\left(\sum_{a:\mu_a < \gamma} \bar{\psi}_a(t) d(\mu_a, \gamma), \sum_{a:\mu_a > \gamma} \bar{\psi}_a(t) d(\mu_a, \gamma)\right)\right) \pm \varepsilon_t\right),$$

which implies

$$\psi_a(t) \leq \beta_t \frac{\Pi_{t-1}(\theta_{t,a} < \gamma < \theta_{t,b})}{\max_{a,b} \Pi_{t-1}(\theta_{t,a} < \gamma < \theta_{t,b})} + (1 - \beta_t) \frac{\Pi_{t-1}(\theta_{t,b} < \gamma < \theta_{t,a})}{\max_{a,b} \Pi_{t-1}(\theta_{t,b} < \gamma < \theta_{t,a})}.$$

$$= \frac{\exp\left(-t \left(\min\left(\sum_{a:\mu_a < \gamma} \bar{\psi}_a(t) d(\mu_a, \gamma), \sum_{a:\mu_a > \gamma} \bar{\psi}_a(t) d(\mu_a, \gamma)\right)\right) - \varepsilon_t\right)}{\max_a \exp\left(-t \left(\min\left(\sum_{a:\mu_a < \gamma} \bar{\psi}_a(t) d(\mu_a, \gamma), \sum_{a:\mu_a > \gamma} \bar{\psi}_a(t) d(\mu_a, \gamma)\right)\right) + \varepsilon_t\right)}$$

$$= \exp\left\{-t \left[\min\left(\sum_{a:\mu_a < \gamma} \bar{\psi}_a(t) d(\mu_a, \gamma), \sum_{a:\mu_a > \gamma} \bar{\psi}_a(t) d(\mu_a, \gamma)\right) \right.\right.$$

$$\left.\left. - \min_a \min\left(\sum_{a:\mu_a < \gamma} \bar{\psi}_a(t) d(\mu_a, \gamma), \sum_{a:\mu_a > \gamma} \bar{\psi}_a(t) d(\mu_a, \gamma)\right)\right] - 2\varepsilon_t\right\}$$

$$\leq \exp\left\{-t \left[\min\left(\sum_{a:\mu_a < \gamma} (w_a^* + c) d(\mu_a, \gamma), \sum_{a:\mu_a > \gamma} (w_a^* + c) d(\mu_a, \gamma)\right) \right.\right.$$

$$\left.\left. - \min\left(\sum_{a:\mu_a < \gamma} w_a^* d(\mu_a, \gamma), \sum_{a:\mu_a > \gamma} w_a^* d(\mu_a, \gamma)\right)\right] - 2\varepsilon_t\right\}.$$

$$\leq \exp\left\{-t \left[c \min\left(\sum_{a:\mu_a < \gamma} d(\mu_a, \gamma), \sum_{a:\mu_a > \gamma} d(\mu_a, \gamma)\right) - 2\varepsilon_t\right]\right\}$$

When $\varepsilon_t \to 0$ the $\psi_a(t)$ is bounded by an exponential decay term, therefore

$$\sum_t \psi_a(t) \mathbb{1}\{\bar{\psi}_a(t) \geq w_a^* + c\} \leq \infty.$$

Therefore, we have $\bar{\psi}(t) \to \boldsymbol{w}^*$, and by the conclusions above, $\boldsymbol{N}(t)/t \to \boldsymbol{w}^*$.

### A.2 $d$-dimensional CHM problem when $d \geq 2$

In this section, we provide further details and discussions about potential extensions of Thompson-CHM algorithm to the higher dimensional case. Before moving forward, we first prove Theorem 5, which gives insight into how to generalize our algorithm.

#### A.2.1 Proofs of Theorem 5

For the infeasible case when $\gamma \notin \mathrm{Conv}(\boldsymbol{\mu})$, the proof is identical to the one-dimensional case and we omit it here.

For the feasible case when $\gamma \in \mathrm{Conv}(\boldsymbol{\mu})$. We assume $\boldsymbol{\lambda}^* \in \mathrm{Alt}(\boldsymbol{\mu})$ is the optimal solution in the game $T^*(\boldsymbol{\mu})^{-1} = \sup_{\boldsymbol{w} \in \Delta} \inf_{\boldsymbol{\lambda} \in \mathrm{Alt}(\boldsymbol{\mu})} \sum_a w_a d(\mu_a, \lambda_a)$. For any $\mu_i \in \{\mu_1, \cdots, \mu_K\} \backslash \mathrm{Vert}(\mathrm{Conv}\{\boldsymbol{\mu}\})$, we consider two different cases.

- Case 1: if $\mu_i \in \mathrm{Conv}(\boldsymbol{\lambda})$, then we have $\lambda_i = \mu_i$, and therefore $w_i = 0$.

- Case 2: if $\mu_i \in \mathrm{Conv}(\boldsymbol{\mu}) \backslash \mathrm{Conv}(\boldsymbol{\lambda})$, in this case, $\lambda_i \neq \mu_i$, WLOG we assume $\mu_1, \mu_2, \cdots, \mu_s$ are the means that differs from those in the optimal solution $\boldsymbol{\lambda}^*$, i.e. $\{1, 2, \cdots, s\} = \{j | \mu_j \neq \lambda_j^*\}$, so $1 \leq i \leq s$. If $w_i \neq 0$, then $d(\mu_i, \lambda_i^*) > d(\mu_j, \lambda_j^*)$ for all $j \in \{1, 2, \cdots, s\} \backslash \{i\}$, this contradicts with the fact that $\mu_i \in \mathrm{Conv}\{\mu_1, \cdots, \mu_s, \lambda_1, \cdots, \lambda_s\}$.

Combining the statements above, we can see that for all $\mu_i$'s that is not one of the vertices of $\mathrm{Conv}\{\boldsymbol{\mu}\}$, in order to win the optimization game $T^*(\boldsymbol{\mu})^{-1}$, no proportion of the corresponding arm should be sampled.

#### A.2.2 Potential extension of Thompson-CHM algorithm

As discussed in Section 6.2, the Thompson-CHM algorithm outperforms the trivial solution (first checking if $\gamma$ is smaller than the minimum mean and then checking if $\gamma$ is larger than the maximum mean) in both generalizability and optimality. For the generalizability part, the Thompson-CHM can possibly generalize to higher dimensional cases and we will discuss more details in this section. For the optimality part, by the results in Kaufmann et al. (2018), the allocations of different arms in the trivial solution are not in align with the optimal $\boldsymbol{w}^*(\boldsymbol{\mu})$ and therefore, lead to a sub-optimal sample complexity compared to the Thompson-CHM algorithm. For example, in the infeasible case when $\gamma < \mu_1 < \cdots < \mu_K$, by utilizing the result in Kaufmann et al. (2018) twice, sampling proportion of arm $K$ is $\frac{2d(\mu_K, \gamma)^{-1}}{\sum_{i=1}^K d(\mu_i, \gamma)^{-1} + d(\mu_K, \gamma)^{-1}}$, and for arm $j$ satisfying $1 \leq j \leq K-1$, its sampling proportion is $\frac{d(\mu_j, \gamma)^{-1}}{\sum_{i=1}^K d(\mu_i, \gamma)^{-1} + d(\mu_K, \gamma)^{-1}}$. This is a direct example of the sub-optimality of the trivial solution to the CHM problem in the one-dimensional case.

Theorem 5 demonstrates an important phenomenon that shares in all dimensions: *in the feasible case, the optimal strategy should only sample arms whose means are extreme points, and in the infeasible case, it should sample all arms.* And if we can prove analogs of the results for the stopping rule, it is possible to fully extend Thompson-CHM algorithm to higher dimensions. We call $\lambda^*$ the point that is on the boundary of $\mathrm{Conv}(\boldsymbol{\mu})$ that minimizes the $l_2$ distance between $\gamma$ and $\gamma^*$, and we vertically project all the means $\mu_1, \cdots, \mu_K$ to the line that connects $\gamma$ and $\gamma^*$, and denote the projected points to be $\mu_1^*, \cdots, \mu_K^*$, then the $d$-dimensional distributions with means $\mu_1, \cdots, \mu_K$ are feasible (infeasible) with respect to $\gamma$ if and only if the 1-dimensional distributions with means $\mu_1^*, \cdots, \mu_K^*$ are feasible (infeasible) with respect to $\gamma$ on the line that connects $\gamma$ and $\gamma^*$. With this important fact, it is possible to prove our conjecture that the analog of Thompson-CHM (described below) is also asymptotically optimal in higher dimensions using similar techniques in the one-dimensional case.

We now generalize the Thompson-CHM algorithm to higher dimensions by replacing the Bernoulli distribution with a categorical distribution with parameters $\beta_i = f_i(\mu_{s_1}, \cdots, \mu_{s_m}, \gamma)$. Assuming oracle access to the functions $f_i$ for $1 \leq i \leq k$, the analog of Thompson-CHM algorithm in $d$-dimensional case is stated below. The future work is to find the exact form of functions $f_i$ and prove the asymptotic optimality of $d$-dimensional Thompson-CHM algorithm.

---

**Algorithm 2** $d$-dimensional Thompson-CHM ($d \geq 2$)

---

**Input:** stopping rule $\tau$ with threshold function $\text{Thresh}(\delta, t)$, risk $\delta$, threshold $\gamma$, Categorical distribution parameter $\beta_1, \cdots, \beta_K$.
**Output:** decision rule $I_\pi(\boldsymbol{\mu}) \in \{\text{feasible}, \text{infeasible}\}$
**for** $t = 1, \cdots$ **do**
  **if** stopping rule $\tau$ holds **then**
    **if** $\tau = \tau_3$ **then**
      **return** $I_\pi(\boldsymbol{\mu}) = \{\text{feasible}\}$
    **else**
      **return** $I_\pi(\boldsymbol{\mu}) = \{\text{infeasible}\}$
    **end if**
  **end if**
  Sample $\boldsymbol{\theta}_t = (\theta_{t,1}, \cdots, \theta_{t,K}) \sim \Pi_{t-1}(\cdot | \boldsymbol{\mu} \text{ feasible})$.
  Sample $B \sim \text{Categorical}(\beta_1, \cdots, \beta_K)$
  **if** $B = i$ **then**
    Play arm $A_t = i$
  **end if**
**end for**

---

