# OpenReview forum: "An Asymptotically Optimal Algorithm for the Convex Hull Membership Problem"
_TMLR — Accepted by TMLR_

### Review · Reviewer_sDAi · 2025-06-27

**Summary Of Contributions:**

This paper studies a fixed-confidence pure exploration problem where one interacts with a $K$-armed bandit environment and aims to test the membership of an input element in the convex hull of the unknown reward vector of arms. This generlaizes the thresholding bandit problem and earlier works on finding the lowest mean in a pure exploration setting.

For one-dimensional rewards, the paper shows an asymptotic lower bound on sample complexity and a mathcing upper bound using a variant of Thompson Sampling. For higher dimensional feedback, they show an analogous lower bound. They also have synthetic experiments validating their theoretical results against the theoretical lower bound.

**Audience:**

Yes

**Claims And Evidence:**

Yes

**Requested Changes:**

See weaknesses above.

## writing suggestions/typos
* The notation can be a bit confusing in writing the decision rule as $I_{\pi}(\mu)$ and then using $\mu$ in Definition 3.2, as here (at least to my understanding) $\mu$ is not a test/input vector of means, but represents the true vector of means generating the sampling data from earlier in your setup. The true input whose membership you are testing for is $\gamma$.
* $kl(\delta,1-\delta)$ should be $KL(\delta,1-\delta)$ in 3rd paragraph of p 4.
* Lemma 5.1 should be Lemma 5.1 in Appendix A.1.
* It should be specified on top of p. 5 that $\textbf{N}(t)$ is the vector of selection counts.
* If I understand correctly, Lemma 5.1 is saying for any Thresh function satisfying the inequality, you'll have asymptotic optimality. This should be made more clear, Thompson-CHM would be desirably introduced before this lemma.
* Also, it seems like Lemma 5.1 is the main result for Thompson-CHM and so it should be a theorem?
* what is $\theta_r$ in Theorem 2? Is this supposed to be $\theta_t$? This notation is also used in Theorem 4.
* The way Algorithm 1 is written appears a bit strange. Why is the unknown mean reward vector $\mu$ an input to the algorithm? How can one compute the posterior without this knowledge?

**Strengths And Weaknesses:**

## strengths
* matching upper and lower bounds on sample complexity for this problem
* a generalized loewr bound for d-dimensional CHM problem which generalizes other pure exploration problems such as thresholding bandits and test for lowest mean.

## weaknesses
* The technical novelty feels a bit limited since much of the core ideas have been previously developed for finding the lowest mean, and the main add-on here is to do a two-arm sampling scheme to account for both boundaries of an interval. But, it remains unclear how to generalize this to higher dimensions and the authors admit it is left for future work.
* The experiments also feel a bit limited. It'd be interesting for instance to compare to the naive/easy strategy mentioned on the bottom of p. 7 or to see how the proposed Thompson-CHM fares for higher-dimensional CHM problems, even if the theory is not yet complete there.

---

> ### Author Response · Authors · 2025-07-29
>
> We deeply thank the reviewer for the thoughtful and detailed feedback on our work, and the effort to help us improve our submission. Based on the identified weaknesses and suggestions, we have made the following corrections and clarifications in the revised manuscript (for your convenience, we used BROWN texts to show the revised content in the revision):
>
> 1. On the technical novelty of our contribution and limitation of experiments: We sincerely thank the reviewer for raising the concern about the degree of technical novelty. While our theoretical tools build upon existing information-theoretic and asymptotic frameworks from best-arm identification, we believe that the technical novelty of our work lies in the synthesis of geometric reformulation and theoretical analysis for the Convex Hull Membership problem. In particular, we introduce a novel geometric view that generalizes the min/max in one-dimensional to extreme points in $d$-dimensional case, and use the insight to guide the design of the algorithm. This leads us to an important structural observation: in the feasible case, the optimal strategy should only sample arms whose means are extreme points, and in the infeasible case, it should sample all arms. Most importantly, we remark that we tackled a major challenge in extending Thompson Sampling to learn extreme means simultaneously. This was inspired from a two-arm sampling construction proposed in the BAI setting by Russo 2016, and to the best of our knowledge, conditional Thompson sampling and two-arm sampling have never been combined before and is novel in technique. Regarding the limitation of experiments, we note that Kaufmann et al. (2018) did not open-source their code, therefore it is challenging to compare the Thompson-CHM to the naive approach directly. Nevertheless, we carefully discussed cases that results in sub-optimality in the naive approach in Section 5 and appendix. To further strengthen the empirical component of our work, we added a new experiment in the revision that studies the behavior of the sample complexity with respect to $\delta$. As shown in Figure 3 in our revision, the average sample complexity grows linearly with $\log(1/\delta)$, empirically supporting the asymptotic optimality of our algorithm.
>
> 2. In response to the reviewer's suggestion regarding the notation of the decision rule, we completely agree with it and we have updated the decision rule notation to $I_{\pi}(\gamma)$ as recommended.
>
> 3. We hope to clarify that in our paper, "kl" and "KL" represents different meanings: "kl" refers specifically to the KL divergence between two binary reward case. To avoid ambiguity, we removed all instances of "KL" in the revision. For more details and discussions, please see Section 4.
>
> 4. We thank the reviewer for pointing out the typo. The typo "lemme" has been corrected to "lemma" in the revision.
>
> 5. In the revision uploaded, we have added a clarification in the text where $N(t)$ is used, indicating that it denotes the vector of selection counts.
>
> 6. We thank the reviewer for the very important comment on Lemma 5.1. To explain why the result is listed as a lemma before the Thompson-CHM algorithm is introduced, this is because that Lemma 5.1 is saying that ensuring the sampling proportions converge to $w^\ast$ is sufficient for reaching the optimal sample complexity in an asymptotic manner. And after Thompson-CHM is introduced, we use Theorem 2 to demonstrate that, with appropriate parameter choices in the algorithm, we can guarantee the sampling proportion converges to $w^\ast$ in Thompson-CHM, and therefore, achieving asymptotic optimality. In the original submission, we stated "therefore Thompson-CHM is $\delta$-correct" in the lemma and it understandably causes confusion. To further clarify the structure, we have revised Lemma 5.1, Theorem 2, Lemma 6.1 and Theorem 4 in the revision to better delineate the logic. While we believe the current structure is more coherent, we acknowledge that Lemma 5.1 could equally be promoted to a theorem, and we are happy to make this change should the reviewer prefer it.
>
> 7. We thank the reviewer for pointing out the typo. We have corrected both $\theta_r$ to $\theta_t$ in Theorem 2 and Theorem 4.
>
> 8. We thank the reviewer for the concern in the algorithm. We have remove $\mu$ from the input. As correctly noted, $\mu$ is the unknown parameter the algorithm seeks to learn and should not be in the input list. To further clarify how the algorithm operates, the posterior in the Thompson Sampling does not require the true $\mu$, and we maintain a posterior distribution over $\mu$ for each arm based on observed data (success and failures). For example, for Beta(1,1) prior, and the observed data has $S_i$ successes in $N_i$ pulls, then the posterior is $\text{Beta}(S_i+1, N_i-S_i+1)$, and we do not need $\mu$ to compute tposterior, instead we observe rewards, update counts and compute the posterior based on those counts.

---

> ### Author Response · Authors · 2025-08-06
>
> We hope to kindly follow up regarding our response and revisions to your comments. Please let us know if there is anything further we can do to address your concerns. We sincerely appreciate your time and feedback again throughout the process.

---

> > ### Comment · Reviewer_sDAi · 2025-08-06
> >
> > Thank you for the detailed response. My main questions about the technical novelty are answered.

---

> > > ### Author Response · Authors · 2025-08-07
> > >
> > > We thank the reviewer again for the helpful comments and time during the review process.

---

### Review · Reviewer_WEyF · 2025-07-06

**Summary Of Contributions:**

The authors of this submission define and tackle the problem of convex hull membership (CHM) in the fixed-confidence pure-exploration bandit setting. In the one-dimensional case---which consumes the bulk of the submission---this amounts to determining, with $1-\delta$ probability, if a given number $\gamma$ lies between the maximum and minimum means of arms; the cost metric is the sample complexity, which is the number of arm pulls to make the decision.

The authors (implicitly) claim that a previous lower bound on best-arm identification by Garivier \& Kaufmann implies a lower bound on CHM; specifically, the authors incorporate the $\gamma$ value and the induced problem's feasibility. The lower bound is phrased in terms of the limit of an algorithm's sample complexity as failure probability $\delta$ approaches 0; optimality is said to be achieved when the limit matches the lower bound.

As the authors discuss in Related Work section, a naive algorithm would take one pass to identify arms whose means lie above $\gamma$ and another to identify those whose means lie below it. But this approach does not reuse measurements between passes. The algorithm proposed by the authors instead toggles between pulling the empirical maximum and pulling the empirical minimum, in a randomized fashion. Aside from being optimal in terms of sample complexity, the allocation of pulls is also optimal.

The authors extend their algorithm to two other settings. First, they generalize from determining point membership to determining interval intersection. Second, they generalize to high-dimensional rewards (and therefore means).

The authors conclude with some experiments that show how close their upper and lower bounds are.

**Audience:**

Yes

**Claims And Evidence:**

No

**Requested Changes:**

See weaknesses. There is not much missing content, but it should be added.

**Strengths And Weaknesses:**

*Strengths*: The problem is a natural extension of prior work; similarly, the authors do a good job establishing their solution in prior solutions. The exposition is generally not hard to follow. And of course, matching upper and lower bounds are a nice addition to the bandit literature.

*Weaknesses*, in descending order of significance:

The authors use the lower bound for best-arm identification to get a lower bound on the CHM, but do not provide an argument as to why this is possible, either in plain English or formal notation.

In the experiments,
- the authors "consider Beta prior" but do not specify which member of the family they chose.
- the authors do not specify their choice of $\delta$ to derive the blue curves in Figure 1.
- the authors do not plot sample complexity for varying $\delta$, even though the analytic results concern limits involving $\delta\to 0$. I would have liked to see how fast or slow the upper bound approaches the lower bound.

The authors should sketch how to derive the posterior $\Pi_{t-1}(\cdot| \mathbf{\mu} ~\mathrm{feasible})$.

Some notation could be cleaned up. For example: $d(\mu_1,\mu_2)$, "kl", and "KL" are all used to denote concepts related to the Kullback-Liebler divergence but it is sometimes challenging to distinguish them from one another.

---

> ### Author Response · Authors · 2025-07-29
>
> We deeply appreciate the reviewer for the insightful comments and suggestions, and the effort to help us improve our submission. The following corrections and clarifications have been made in the revised manuscript in response (for your convenience, we used BLUE texts to show the revised content in the revision):
>
> 1. Regarding using the lower bound of best-arm identification on the CHM problem, we thank the reviewer for raising this important point. While the lower bound in Theorem 1 of Garivier and Kaufmann (2016) was originally developed for Best Arm Identification (BAI), the bound itself—derived from Lemma 1 of Kaufmann et al. (2015)—is based on a general information-theoretic inequality that holds for any $\delta$-PAC algorithm over stochastic bandit models. The theorem and its intuitions rely solely on likelihood ratio inequalities and KL divergence between probability measures, under the assumption that the arm distributions are mutually absolutely continuous, and does not rely on the BAI settings. Consequently, this lower bound applies more generally to any fixed-confidence identification problem with multiple arms and a well-defined notion of correctness in the recent MAB literature, including our CHM setting.
>
> 2. In the experiments, we used Beta(1,1) as the prior. This detail has now been clarified in Section 7 of the revision.
>
> 3. We also clarified that the experiments were conducted using $\delta = \exp(-3)$, as now stated in Section 7.
>
> 4. We appreciate the reviewer for proposing this very important experiment setting. We have incorporated a new experiment in the revision that examines the behavior of the sample complexity with respect to $\delta$. To improve interpretability, the x-axis is represented as $\log(1/\delta)$, and both feasible and infeasible cases are considered. As shown in Figure 3 in our revision, the average empirical sample complexity grows linearly with $\log(1/\delta)$, empirically supporting the asymptotically optimality of our algorithm.
>
> 5. We thank the reviewer for the insightful suggestion on posterior derivation. To clarify, in our theoretical formulation $\Pi_{t-1}(\cdot|\mu \text{ feasible})$ denotes the posterior distribution over the parameter space $\mu$ given the observations up to round $t-1$, conditioned on the event that the current hypothesis is true (feasible). Assume a prior distribution $\Pi_0$ (usually Beta or uniform), we can compute the posterior using the Bayes' rule. To implement the conditioning, we adopt reject sampling: we repeatedly sample from the unconditioned posterior until a sample satisfying feasibility is obtained. We have added some sentences to carefully explain how to derive this posterior in the third paragraph of Section 5.2 in page 6 of the revision.
>
> 6. We thank the reviewer for the helpful comment on KL divergence notations. In order to avoid the abuse of notation, we removed the use of "KL" in our paper to prevent ambiguity. For the rest two, the lowercase "kl" now refers specifically to the KL divergence between two binary reward case, and the function $d(\cdot,\cdot)$ is used for KL divergence between two distributions from exponential families. As discussed in Section 3, for the canonical exponential family, it induces a bijection between the natural parameter and the mean parameter, therefore we can formulate the KL divergence in this way. We kindly hope to retain these two notations which are consistent with the well-known paper in the literature (https://arxiv.org/pdf/1602.04589).

---

> > ### Comment · Reviewer_WEyF · 2025-07-31
> >
> > Thank you for the clarifications!

---

> > > ### Author Response · Authors · 2025-08-01
> > >
> > > We thank the reviewer again for the constructive feedback and time throughout the review process.

---

### Review · Reviewer_qY65 · 2025-07-23

**Summary Of Contributions:**

This paper studies the one-dimensional stochastic convex hull membership problem, in which one is given a constant \gamma and seeks to determine whether \gamma lies within the convex hull (in this case, the line segment) formed by a set of unknown means \mu_1, ..., \mu_K, each corresponding to an unknown distribution. The main idea is to reformulate the problem as a best-arm identification task in the MAB framework, treating each distribution as an arm, and determining whether \gamma lies between the two extreme means.

**Audience:**

Yes

**Claims And Evidence:**

Yes

**Requested Changes:**

Please see above.

**Strengths And Weaknesses:**

Comments/Suggestions:
--
Firstly, I usually do not have many comments regarding the presentation during the review process, but I would like to respectfully note that the overall presentation of the paper needs improvement. I noticed many typos and notation errors throughout the manuscript, including several on the first page. I encourage the authors to carefully proofread the paper. I hope the following comments and suggestions will help improve the clarity and quality of the manuscript.

On page 1, paragraph 3, the expression argmin_{\theta \in theta} (F_1(\theta), ..., F_m(\theta)) is problematic. The notation \theta \in theta should be corrected to \theta \in \Theta, where \Theta denotes the parameter set. Moreover, the meaning of “argmin” in the context of a multi-objective optimization problem is unclear and should be defined more rigorously. I also do not understand why the condition \sum \lambda_i \nabla F(\theta^*) = 0 is claimed to characterize all Pareto-optimal solutions; this point requires further explanation. In addition, it would improve clarity to include curly braces in the hypotheses H_0 and H_1. Lastly, in line 5 of paragraph 3, the notation Conv( {}} appears to contain a formatting error and should be corrected to Conv({}).

On page 4, paragraph 3, the expression argmax_{1 \leq k \leq K} \mu_i should be corrected to argmax_{1 \leq i \leq K} \mu_i.

In Definition 3.2, there is a mismatch in notation: \boldsymbol{\mu} is the mean vector, while \mathcal{D} represents the set of distributions. Therefore, it should not be written as \boldsymbol{\mu} \in \mathcal{D}^K; instead, the correct expression should likely be \nu \in \mathcal{D}^K.

On page 14, in the line immediately preceding section A.1.2, the index i should range over the set {1, ..., K}, so i \in \boldsymbol{\mu} should be corrected to i \in \{1, ..., K\}.

Since the case of d \geq 2 is only discussed in the final subsection of the appendix, I suggest the authors explicitly define the problem as one-dimensional in the main text to improve clarity. Moreover, the proposed method appears inherently limited to the one-dimensional setting, and any extension to higher dimensions (d \geq 2) would likely be highly inefficient, as the number of extreme points can grow exponentially with dimension.

In the introduction, line 2, the phrase “has make” should be corrected to “has to make.”

Regarding the structure of the introduction, the authors discuss the multi-armed bandit (MAB) problem in the first paragraph and the convex hull membership (CHM) problem in the second, without clearly establishing the connection between them. As someone familiar with MAB but not with CHM, I found the transition unclear. It is only in Section 3 that the reader understands that each distribution is treated as an arm in the MAB framework. It would help if the second paragraph of the introduction explained more clearly how the CHM problem is modeled as a best-arm identification problem within the MAB framework.

In Section 4, paragraph 2, it is not clear why \boldsymbol{\lambda} is treated as a MAB model. From the appendix, it seems to be simply a vector. This should be clarified.

In Section 5.1, please define the notations N^- and N^+ explicitly.

Strengths and Weakness
---
Overall, I find the problem interesting and relevant to the TMLR audience. While the proposed approach and proofs are extensions of existing work (notably Kaufmann et al., 2018), they appear sound and well-motivated. Some limitations of the method include its restriction to the one-dimensional case and the focus on asymptotic guarantees. However, I do not consider these to be significant weaknesses for a TMLR submission. Therefore, I support acceptance of the paper, provided the authors substantially improve its presentation in response to the above comments.





Minor Questions
--
When defining the decision rule I_{\pi}(\mu) on page 4, should it not also be a function of \gamma?

In Section 3, the authors assume that \gamma is not equal to any of \mu_1 through \mu_K. While it is clear that having \gamma equal to one of the extreme values causes difficulties, why is it problematic when \gamma = \mu_2 or \mu_{K-1}?

Are there any assumptions made about the distributions \nu_1, ..., \nu_K, such as being sub-Gaussian?

---

> ### Author Response · Authors · 2025-07-29
>
> We sincerely thank the reviewer for the very constructive feedback and suggestions, and the effort to help us improve our submission. Based on the comments, we made the following corrections and clarifications in the revised manuscript (for your convenience, we used RED texts to show the revised content in the revision):
>
> 1. On page 1, paragraph 3: we appreciate the reviewer’s comment regarding the use of "argmin" and the formulation of the multi-objective setting. To address this, we have rewritten this part to more rigorously explain how the CHM problem discussed in our paper connects to the Pareto optimality framework. The revised version now formally defined Pareto optimality, and avoided using "argmin" and hypothesis testing for improved clarity. Regarding the condition $\sum \lambda_i \nabla F(\theta^*) = 0$, it is the first-order necessary and sufficient condition for Pareto optimality. Intuitively, Pareto optimality means no direction simultaneously decreases all objectives, i.e., the negative orthant is not contained in the convex hull of the gradients, which leads to this condition. Additionally, we have corrected the formatting of Conv({}) in the same paragraph.
>
> 2. On page 4, paragraph 3: we have corrected two expressions to $argmax_{1 \leq i \leq K} \mu_i$ in the revision.
>
> 3. In Definition 3.2: we have updated the notation to $ \nu \in \mathcal{D}^K$.
>
> 4. On page 14 (page 15 in revision): in the line preceding section A.1.2, we have corrected the set i range over to $\{1,...,K\}$.
>
> 5. We appreciate the reviewer's thoughtful comments regarding the case of $d \geq 2$ being only discussed in the final subsection of the appendix. It is true that our theoretical and algorithmic approaches in the main text focuses on the one-dimensional case, we would like to deliver the message that the proposed formulation and geometric perspective are not inherently one-dimensional: one of our contribution is posing the CHM problem as a formal pure exploration exploration problem, and the mathematical formulation makes sense in all dimensions and generalization of our idea to $d$-dimensional case is natural, and we discuss extensions and challenges in higher dimensions in the appendix. Although the number of extreme points may grow exponentially in $d$, according to Caratheodory' theorem, the test point can always be written as convex combination of at most $d+1$ points. So the asymptotic sample complexity might not depend on all extreme points but only ones needed to represent the test point. Additionally, our approach generalizes the existing min/max concepts restricted to 1-dimensional case to extreme points in higher dimensions and developed the idea to sample from extreme points from a categorical distribution, and opens the door to algorithmic strategies that exploit geometric structure and sparsity in practice. Following our generalization idea, we also proved a lower bound for $d$-dimensional case (Theorem 5) and demonstrated an important phenomenon that works in all dimensions (in the feasible case, the optimal strategy should only sample arms whose means are extreme points, and in the infeasible case, it should sample all arms) discussed in the appendix. To clarify the current scope, we have explicitly stated that the focus of this work is the one-dimensional case in the abstract and the summary of our work at the end of the introduction, and we are happy to emphasize this further per the reviewer's suggestion.
>
> 6. In the introduction, line 2: we updated "has make" to "is required to make".
>
> 7. We thank the reviewer's helpful comment on the structure of the introduction. To address this, we rewrote the second paragraph of the introduction to better explain how CHM is formulated as a MAB problem. This improves the transition from the general MAB context to our specific CHM setting.
>
> 8. In section 4, paragraph 2: we have simplified the notation by just saying "We define $Alt(\mu)$ to be the set of bandit models where the identification result is different from that in $\mu$", and this can avoid the confusion regarding $\lambda$.
>
> 9. In section 5.1: we thank the reviewer for pointing out $N^-$ and $N^+$, they were a typo and are now consistently written as $\mathbb N^+$.
>
> 10. For the decision rule, we thank the reviewer for the helpful comment. We have updated the decision rule notation to $I_{\pi}(\gamma)$ for clarity in the revision.
>
> 11. We agree that it suffices to distinguish between $\gamma$ and the two extreme values in 1-dimensional case. In an earlier draft, we assumed $\mu_1 \leq \mu_2 \leq \cdots \leq \mu_K$ so we need $\gamma$ to be different from all means to avoid edge cases. We have updated the assumption in Section 3 in the revision so under the current assumption $\mu_1 < \mu_2 \leq \cdots \leq \mu_{K-1}< \mu_K$, $\gamma$ only needs to differ from $\mu_1$ and $\mu_K$.
>
> 12. The distributions $\nu_1,\cdots,\nu_K$ are assumed to be in the exponential family as stated in Section 3.

---

> > ### Comment · Reviewer_qY65 · 2025-07-30
> > **Response**
> >
> > Thank you for the detailed reply and revision, and I do not have further questions.

---

> > > ### Author Response · Authors · 2025-07-30
> > >
> > > We thank the reviewer again for the thoughtful comments and time during the review process.

---

### Decision · Action_Editor_VmtG · 2025-09-05

**Recommendation:** Accept as is

**Audience:**

Yes

**Audience Explanation:**

The problem of convex hull membership (CHM) in a stochastic bandit setting will be of interest to members of the TMLR community that work on bandit algorithms, particularly from a theoretical point of view.

**Claims And Evidence:**

Yes

**Claims Explanation:**

The paper studies the problem of convex hull membership (CHM) in a stochastic bandit setting. The paper focuses on the one-dimensional setting (so the convex hull is a segment between the two extreme means). The authors propose a best-arm identification-type algorithm that toggles between pulling the empirical maximum and empirical minimum, and show its optimality by comparing with a previously established lower bound. After some clarifications from the authors, all reviewers agree that the results and experiments seem correct and sound.